# Proteasomal degradation of the tumour suppressor FBW7 requires branched ubiquitylation by TRIP12

Omar M. Khan [1,2✉], Jorge Almagro [1,9], Jessica K. Nelson[1,9], Stuart Horswell [3], Vesela Encheva[4], Kripa S. Keyan[2], Bruce E. Clurman[5], Ambrosius P. Snijders [4] & Axel Behrens [1,6,7,8✉]

The tumour suppressor FBW7 is a substrate adaptor for the E3 ubiquitin ligase complex SKP1-CUL1-F-box (SCF), that targets several oncoproteins for proteasomal degradation. *FBW7* is widely mutated and FBW7 protein levels are commonly downregulated in cancer. Here, using an shRNA library screen, we identify the HECT-domain E3 ubiquitin ligase TRIP12 as a negative regulator of FBW7 stability. We find that SCF[FBW7]-mediated ubiquitylation of FBW7 occurs preferentially on K404 and K412, but is not sufficient for its proteasomal degradation, and in addition requires TRIP12-mediated branched K11-linked ubiquitylation. *TRIP12* inactivation causes FBW7 protein accumulation and increased proteasomal degradation of the SCF[FBW7] substrate Myeloid Leukemia 1 (MCL1), and sensitizes cancer cells to anti-tubulin chemotherapy. Concomitant *FBW7* inactivation rescues the effects of TRIP12 deficiency, confirming FBW7 as an essential mediator of TRIP12 function. This work reveals an unexpected complexity of FBW7 ubiquitylation, and highlights branched ubiquitylation as an important signalling mechanism regulating protein stability.

[1] Adult Stem Cell Laboratory, The Francis Crick Institute, London, UK. [2] Hamad Bin Khalifa University, College of Health and Life Sciences Qatar Foundation, Education City, Education City, Doha, Qatar. [3] Bioinformatics and Biostatistics, London, UK. [4] Proteomics, The Francis Crick Institute, London, UK. [5] Fred Hutchinson Cancer Research Center, Seattle, WA, USA. [6] Cancer Stem Cell Laboratory, Institute of Cancer Research, London, UK. [7] Division of Cancer, Department of Surgery and Cancer, Imperial College London, London, UK. [8] Convergence Science Centre, Imperial College London, London, UK. [9] These authors contributed equally: Jorge Almagro, Jessica K. Nelson. ✉email: okhan@hbku.edu.qa; axel.behrens@icr.ac.uk

SCF[FBW7] is a widely studied E3 ubiquitin ligase complex of the Cullin-RING family that targets many well-characterised oncoproteins including c-MYC, CyclinE, MCL1, and Notch intracellular domain 1 (NICD1) for ubiquitylation-mediated proteasomal degradation[1]. Critical to SCF[FBW7] activity is the substrate adaptor FBW7 that binds to its substrates in a phosphorylation-dependent manner[2]. Loss of FBW7 function results in the accumulation of these proteins which result in increased tumorigenesis and chemotherapy resistance[1,3–7]. Consequently, *FBW7* is the most mutated ubiquitin-proteasome pathway gene in human cancers[8]. SCF[FBW7], like many other E3 ubiquitin ligases, undergoes efficient autoubiquitylation[9,10]. We previously showed that the deubiquitinase USP9X inhibits FBW7 degradation by deubiquitylating FBW7, resulting in protein stabilisation, providing a molecular explanation for the frequent USP9X mutations in colorectal cancer[11]. Thus, regulation of FBW7 stability is a mechanism that contributes to tumour formation.

The thyroid hormone receptor interactor 12 (TRIP12), also known as the E3 ubiquitin ligase for Arf (ULF), is a HECT-domain E3-ligase. The *yeast* protein Ufd4 was identified in a screen to extend polyubiquitin chains on a protein substrate that is fused with a ubiquitin moiety on its N-terminus, which targets it for proteasomal degradation[12], and subsequent work identified TRIP12 as the homologous protein to function in the ubiquitin fusion degradation pathway in mammalian cells[13]. Additionally, TRIP12 responds to DNA damage and oncogenic stress by targeting RNF168 and p19ARF for proteolysis[14,15].

Ubiquitin moieties can be conjugated through one of their lysine residues (K6, K11, K27, K29, K33, K48 and K63) or the N-terminal methionine residue (M1). Homotypic ubiquitin chains comprise only a single-linkage type whereas heterotypic chains contain mixed linkages within the same polymer. Thus, the system offers versatility to assemble a large variety of different ubiquitin polymers[16]. Whereas the roles of homotypic poly-ubiquitylation are relatively well-characterised, the targets and functions of heterotypic polyubiquitylation are just beginning to be uncovered[17].

Here using an FBW7 protein stability screen we identify TRIP12 as a negative regulator of FBW7 protein stability. Genetic deletion of *TRIP12* stabilised FBW7 and enhanced FBW7-mediated degradation of its substrate MCL1, rendering cancer cells sensitive to the chemotherapy drug Taxol. Importantly, SCF[FBW7] autoubiquitylation on two FBW7 lysine residues, K404 and K412, facilitated binding to TRIP12 and this interaction promoted K11-linked ubiquitin branching on FBW7, leading to its proteasomal degradation. Thus, SCF[FBW7]-mediated autoubiquitylation is not sufficient for efficient proteasomal degradation of FBW7, but rather functions as a priming event that in combination with subsequent K11-linked branching by TRIP12 culminates in proteasomal degradation of FBW7.

## Results

**An shRNA library screen identifies post-translational regulators of FBW7 stability**. To identify regulators of FBW7 protein stability, we used a previously published experimental strategy[18]. We generated a bi-cistronic construct expressing a DsRed fluorescent protein followed by an internal ribosome entry site (IRES) and the FBW7α open reading frame fused at the N-terminus to EGFP. The DsRed protein serves as an internal control for EGFP-FBW7α protein stability in response to genetic or pharmacological perturbations (DsRed-IRES-EGFP-FBW7α; Fig. 1a). We generated a stable HEK293T DsRed-IRES-EGFP-FBW7α reporter cell line. EGFP-FBW7α could be detected by both GFP and FBW7α-specific western blot antibodies (Supplementary Fig. 1a). Additionally, EGFP-FBW7α interacted with

endogenous E3-RING ligase complex partners CUL1 and SKP1 (Supplementary Fig. 1b), suggesting that the fusion protein has retained the ability to interact with SCF-complex protein partners. We used the EGFP-FBW7α/DsRed ratio as a readout for FBW7 protein stability. Proteasome inhibitor (MG132) treatment stabilised EGFP-FBW7α leading to an increase in the EGFP-FBW7α/DsRed ratio (Supplementary Fig. 1c, d). Conversely, a protein translation inhibitor, cycloheximide, led to a decrease in the EGFP-FBW7α/DsRed ratio (Supplementary Fig. 1e).

We transduced DsRed-IRES-EGFP-FBW7α reporter cells with an shRNA library targeting ~5500 druggable genes, where each gene was targeted by at least five shRNAs in a pool of ~27,500 shRNAs (Fig. 1b). Transduced cells were sorted into two distinct populations, based on low and high EGFP-FBW7α/DsRed ratio (Supplementary Fig. 1c). Enriched hits identified after deep sequencing and DNA barcode deconvolution analysis were the most common genes from the ubiquitin-proteasome pathway (Fig. 1c). Depletion of *CSN5*, a critical subunit of the COP9 signalosome de-neddylation complex that inactivates Cullin-RING ligases, and depletion of *FBW7* itself, led to the largest reduction in EGFP-FBW7α protein levels (Fig. 1d). In contrast, components of the proteasome including several members of the *PSMA* and *PSMB* family, and components of SCF[FBW7] and neddylation pathway genes such as *CUL1*, *RBX1, NEDD8*, and *UBE2M2*, were highly enriched as negative regulators of FBW7 stability (Fig. 1d and Supplementary Fig. 1f). Additionally, we identified a HECT-domain E3-ligase, TRIP12, as a negative regulator of FBW7 stability (Fig. 1d). We confirmed the effects of selected library hits, including TRIP12, on epitope-tagged FBW7α stability using two independent siRNAs via western blot and FACS analysis (Fig. 1e, f).

**TRIP12 negatively regulates FBW7 protein stability**. Knockdown of *TRIP12* increased the amount of endogenous FBW7α protein (Fig. 2a) without affecting *FBW7* mRNA levels (Fig. 2b), suggesting that TRIP12 regulates FBW7 protein in a post-translational manner. *TRIP12* depletion only stabilised FBW7α isoform but not the β or γ isoforms (Supplementary Fig. 2a). CRISPR/Cas9-mediated *TRIP12* deletion also stabilised ectopically expressed FLAG-FBW7 protein (Fig. 2c), whereas *TRIP12* mRNA overexpression from the endogenous *TRIP12* locus using CRISPR/Cas9-guided synergistic activation mediator (*TRIP12* SAM) reduced FBW7α protein (Fig. 2d, e). Thus, TRIP12 is a rate-limiting regulator of FBW7 protein levels.

Cycloheximide-chase experiments revealed that *TRIP12* depletion increased the stability of both FLAG-tagged and endogenous FBW7 protein (Fig. 2f, g and Supplementary Fig. 2b). In contrast, *TRIP12* knockdown did not affect the protein stability of the FBW7 F-box deleted mutant (ΔFbox), a FBW7 mutant that cannot be autoubiquitylated by SCF[FBW7] because it lacks the critical F-box domain required for interaction with the Cullin-RING ligase complex (Fig. 2f, g). This result suggested that TRIP12-mediated negative regulation of FBW7 requires SCF[FBW7]-mediated autoubiquitylation. Consistently, FBW7 ubiquitylation was significantly reduced, but not abolished, in *TRIP12*-knockdown cells treated with a proteasome inhibitor (MG132) whereas the FBW7 ΔFbox-mutant was largely unmodified (Fig. 2h). On the contrary, we found increased ubiquitylated FBW7 in *TRIP12*-knockdown cells in the absence of MG132 (Supplementary Fig. 2c). These results suggest that FBW7 is efficiently autoubiquitylated in the absence of TRIP12 but probably not efficiently degraded by the proteasome.

**TRIP12 regulates chemotherapy resistance via MCL1 and FBW7**. We next investigated the effect of *TRIP12* depletion on SCF[FBW7] substrates. There was a reduction in the protein levels of

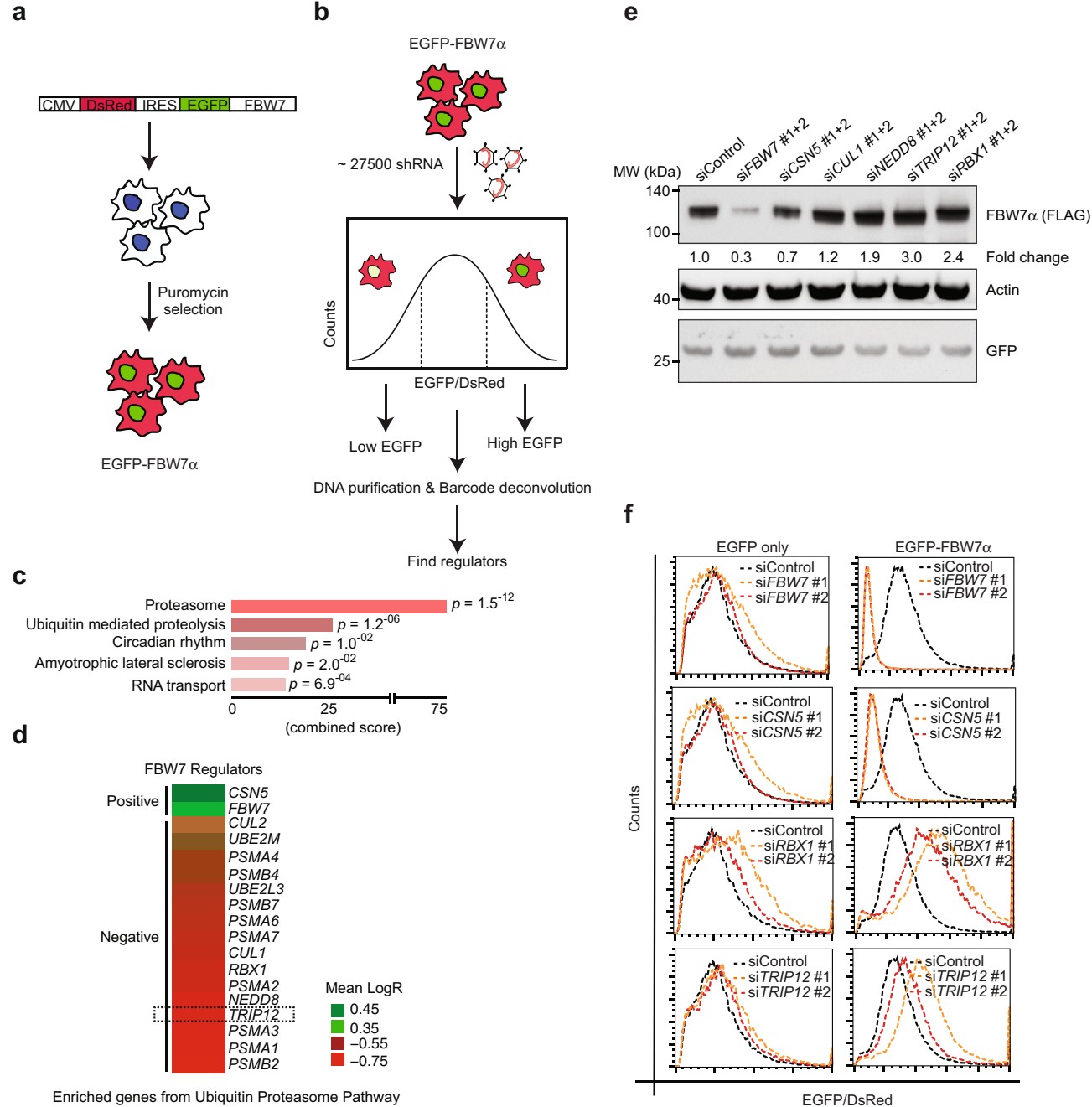

**Fig. 1 A shRNA library screen identifies post-translational regulators of FBW7 stability. a** Schematic of DsRed-IRES-EGFP-FBW7α vector and creation of stable HEK293T cell line. **b** Schematic of pooled shRNA library EGFP-FBW7α stability screen. **c** Pathway analysis of enriched hits showing "Proteasome" and "Ubiquitin-mediated proteolysis" as top enriched gene sets in the screen. Statistics were done by Fisher's exact t-test and adjusted for multiple comparisons. **d** Heat map showing mean logR enrichment values for the shRNAs of each gene from the ubiquitin-proteasome pathway. **e** Western blot validation of selected hits from **d** in HEK293T cells. Numbers represent quantification of FLAG-FBW7 band normalised to GFP loading control and are shown as fold change. The experiment was repeated at least three times with similar results. **f** Flow cytometry validation of two positive and two negative regulators of FBW7 protein stability shown as EGFP-FBW7/DsRed ratio, n = 3 independent experiments. See also Supplementary Fig. 1. Source data are provided as a Source Data file.

MCL1 and CyclinE (Fig. 3a) with no reduction in the corresponding mRNA (Supplementary Fig. 3a), while c-MYC levels were largely unaffected (Fig. 3a and Supplementary Fig. 3a).

In contrast, MCL1 protein was stabilised in *TRIP12*-SAM cells overexpressing endogenous *TRIP12* (Fig. 3b). MCL1 levels were restored by the proteasome inhibitor MG132 in *TRIP12*-knockdown cells, suggesting that MCL1 protein is degraded more efficiently by the proteasome in the absence of *TRIP12* (Fig. 3c).

Inhibition of *FBW7* expression with either siRNA in HEK293T cells, or genetic knockout via CRISPR/Cas9 in HCT116 colon cancer cells, completely rescued MCL1 levels in *TRIP12*-silenced and *TRIP12*-knockout cells, respectively (Fig. 3d and Supplementary Fig. 3b). Thus, MCL1 is stabilised by TRIP12 via negative regulation of FBW7.

Cancer-associated *FBW7* mutations cause chemotherapy resistance via accumulation of MCL1 protein[6]. Genetic deletion

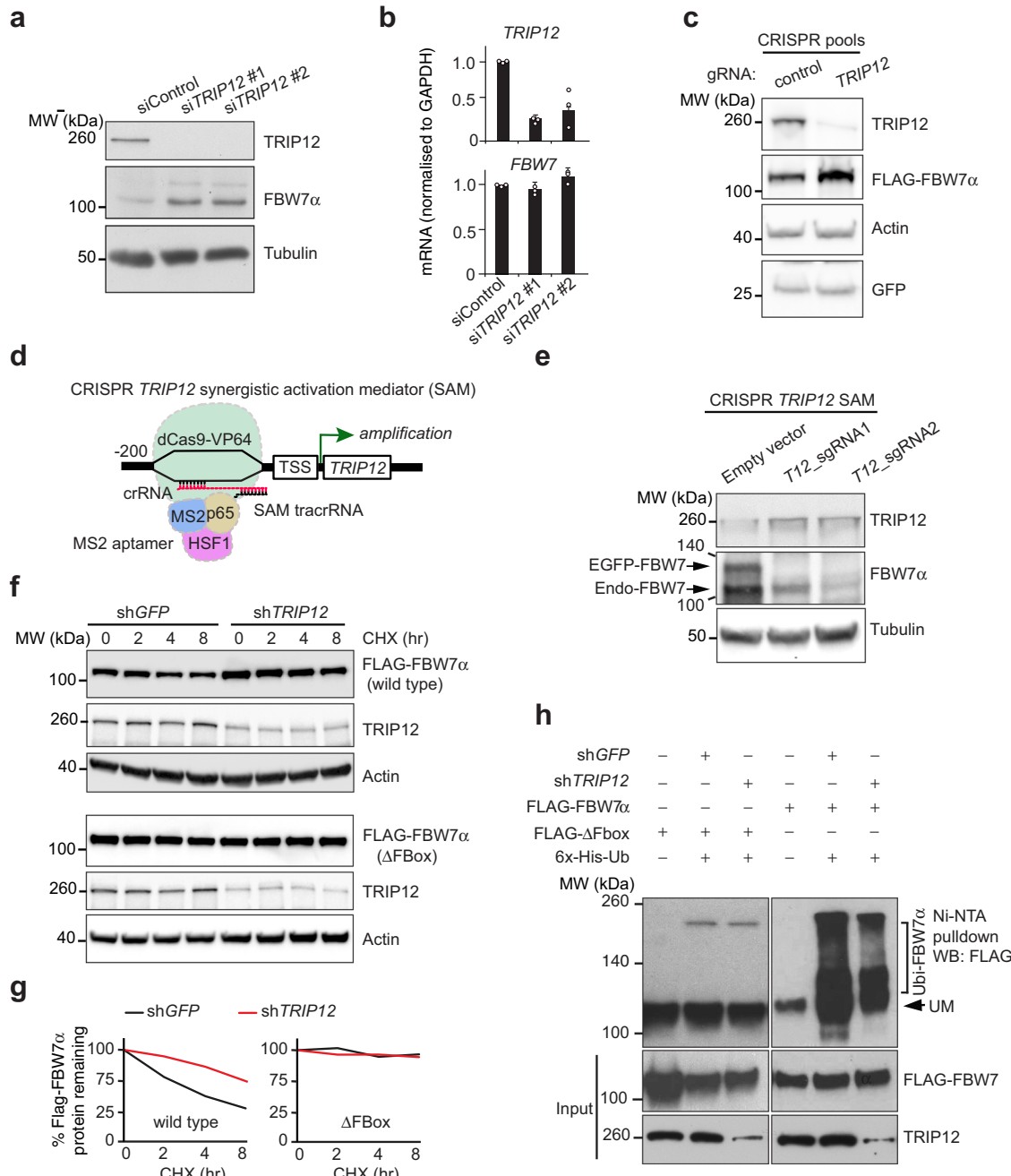

**Fig. 2 TRIP12 negatively regulates FBW7 protein stability. a** Western blot validation of TRIP12 as a negative regulator of endogenous FBW7 on lysates from cells transfected with two independent siRNAs targeting *TRIP12* gene compared to a non-targeting control. **b** qRT-PCR analysis of *TRIP12* and *FBW7*, *n* = 3 independent experiments. Bar graphs represent the mean ± SD of three independent experiments. **c** Western blots for the indicated proteins showing stabilisation of FBW7 in CRISPR/Cas9 HCT116 pools targeting *TRIP12* compared to wildtype parental cells. GFP plasmid was used as transfection control and subsequently GFP and Actin blots were used as loading controls. **d** Schematic of CRISPR/Cas9 SAM sgRNAs targeting endogenous *TRIP12* locus. **e** Western blot validation of *TRIP12* SAM activation and negative regulation of FBW7 in HEK293T cells expressing EGFP-FBW7α. **f** Protein stability of epitope-tagged FBW7 in stable cell lines expressing a lentivirus-mediated GFP-targeting control (sh*GFP*) versus sh*TRIP12*; quantification of FBW7 protein levels normalised to actin from two independent experiments is shown in **g**. **h** Ni-NTA pulldown of ΔFbox-mutant and ubiquitylated wildtype FBW7 in stable HEK293T cells expressing the indicated shRNAs. Western blots in all panels are representative of at least three independent experiments unless otherwise stated. Source data are provided as a Source Data file.

of *TRIP12* had no effect on HCT116 cell proliferation (Fig. 3e) but sensitised those cells to the anti-mitotic chemotherapy Taxol (Fig. 3f). This effect was completely blocked by the concomitant deletion of *FBW7* in *TRIP12*-knockout cells (Fig. 3f). Moreover, *TRIP12* knockdown also sensitised HCT15 and HT29 cells, two additional colorectal cancer cell lines that harbour a wildtype *FBW7* allele, to Taxol, whereas SW837 and SNU175 cells, which harbour a somatic deletion of *FBW7 (L403fs)* and a missense mutation *(R465C)*, respectively, were largely resistant to Taxol regardless of *TRIP12* deletion (Fig. 3g). Thus, TRIP12 modulates Taxol sensitivity through its regulation of FBW7.

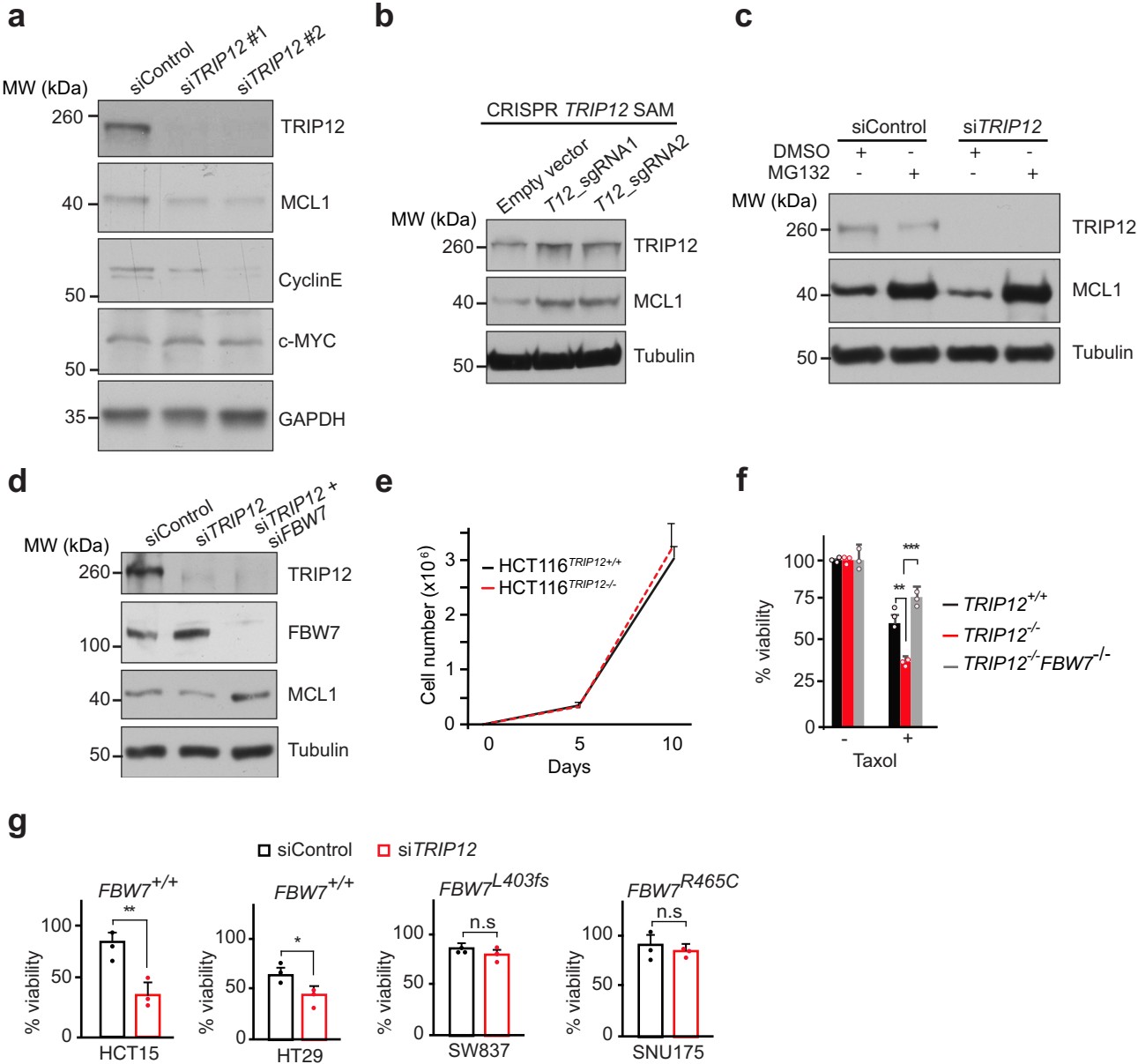

**Fig. 3 TRIP12 regulates chemotherapy resistance via MCL1 and FBW7. a** Western blots showing levels of the indicated FBW7 substrates in HEK293T cells treated with a control siRNA or two independent siRNAs targeting *TRIP12*. **b** Western blots for TRIP12 and MCL1 in cells targeted with CRISPR/Cas9 SAM *TRIP12* sgRNAs or an empty vector control. Tubulin used as a loading control. **c** Western blots for MCL1 from HEK293T cells treated with control and *TRIP12* siRNA ± MG132. Tubulin used as a loading control. **d** Western blot for the indicated proteins in HEK293T cells treated with the indicated siRNAs. Blots in **a–d** are representative of at least three independent experiments. **e** Cell proliferation of HCT116 cells of indicated genotypes, as judged by cell counting on indicated days. Graph shows mean ± SD of three independent experiments. **f** Viability of HCT116 cells of the indicated genotypes as judged by CellTiter-Blue® viability assay in the presence or absence of 100 nM Taxol and shown as % viability relative to the untreated control. Bar graphs represent Mean ± SD of three independent experiments, **$p = 0.004$ and ***$p = 0.0001$. **g** Viability of colorectal cancer cell lines with the indicated genotypes, treated with 100 nM Taxol for 72 h as judged by CellTiter-Blue® assay and shown as % viability relative to the control. Bar graphs represent mean ± SD of three independent experiments, **$p = 0.01$ and *$p = 0.04$, n.s., not significant. *p*-values calculated by two-tailed type two Student's *t*-test. See also Supplementary Fig. 3. Source data are provided as a Source Data file.

**FBW7 autoubiquitylation enhances interaction with TRIP12.** To understand the mechanism of FBW7's negative regulation by TRIP12, we assessed a potential physical interaction of these proteins. Immunoprecipitation (IP) of endogenous FBW7 (Supplementary Fig. 4a) revealed a physical interaction with endogenous TRIP12 (Fig. 4a, lane 3). Interestingly, IP of endogenous TRIP12 showed preferential binding to an FBW7 species of higher molecular weight (Fig. 4a, lane 4). To investigate whether the high-molecular-weight FBW7 species represents a ubiquitylated

form, we established an in vitro system which efficiently catalysed SCF^FBW7 autoubiquitylation (Fig. 4b). Autoubiquitylated FBW7 migrated at a similar molecular weight as the endogenous FBW7 species that interacted with TRIP12 (Fig. 4c). Generation of this high-molecular-weight FBW7 species depended on the presence of ubiquitin and the E1 ubiquitin-activating enzyme in the in vitro ubiquitylation reaction, and was removed by treatment with the promiscuous DUB USP2 (Fig. 4c, d), indicating that high-molecular-weight FBW7 is generated by ubiquitylation (thus

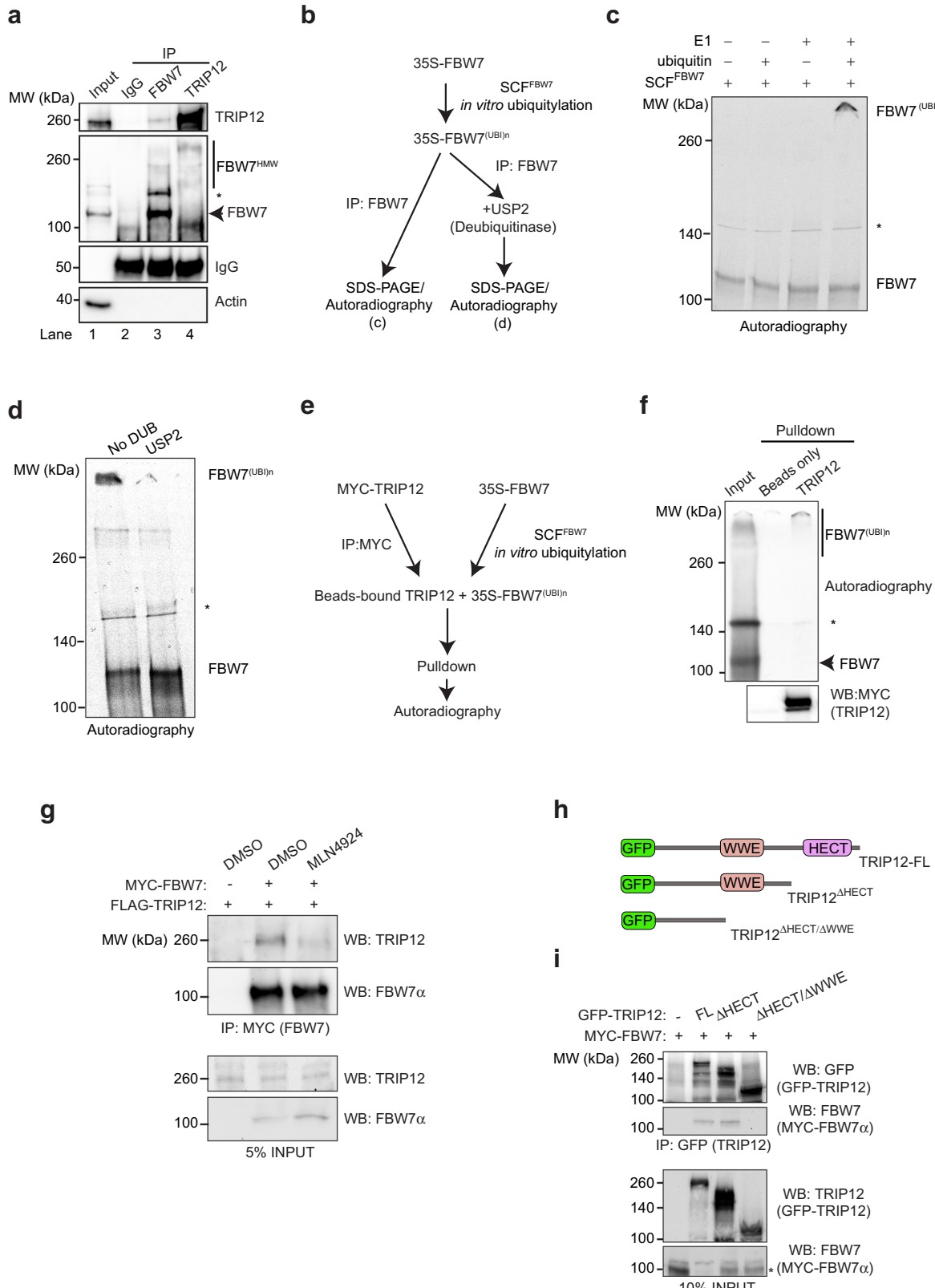

designed FBW7$^{(UBI)n}$. Pulldown experiments using in vitro autoubiquitylated FBW7 confirmed strong preferential binding of TRIP12 to ubiquitylated FBW7 (Fig. 4e, f). Indeed, treating cells with MLN4294, an inhibitor of Cullin neddylation which is required for Cullin-RING ubiquitin ligases (CRLs) activity, stabilised FBW7 protein levels and greatly reduced, but did not abolish,

FBW7 binding to TRIP12 (Supplementary Fig. 4b and Fig. 4g), confirming the requirement of autoubiquitylation for efficient FBW7 and TRIP12 interaction. To identify a potential ubiquitin-binding domain (UBD) in TRIP12, we incubated purified full-length beads-bound TRIP12 with unanchored di-ubiquitins of known topology and detected the loss of di-ubiquitins in the

**Fig. 4 TRIP12 preferentially interacts with ubiquitylated FBW7. a** Co-IP confirming endogenous FBW7 interaction with TRIP12 and vice versa in HEK293T cells, black star (*) represents a nonspecific band of unknown identity. **b** Schematic for experiments in **c** and **d**. **c** In vitro 35S-FBW7α ubiquitylation experiment in the presence of the indicated reagents. **d** In vitro deubiquitylation of 35S-FBW7α by USP2. **e** Schematic of in vitro translated 35S-FBW7 pulldown with MYC-TRIP12 from HEK293T cells. **f** Autoradiograph showing MYC-TRIP12 interaction with ubiquitylated FBW7. **g** Western blots showing the interaction of MYC-FBW7 with FLAG-TRIP12 in HCT116 cells pre-treated for 24 h with either NAE1 inhibitor (MLN4924) or DMSO as control. **h** Schematic of GFP-TRIP12 deletion mutants. **i** Western blots showing the interaction of MYC-FBW7 with GFP-TRIP12, GFP-TRIP12$^{\Delta HECT}$, GFP-TRIP12$^{\Delta HECT/\ \Delta WWE}$ mutants in HCT116 cells, black star (*) represents a nonspecific band of unknown identity. All blots are representative of at least three independent experiments. See also Supplementary Fig. 4. Source data are provided as a Source Data file.

flow-through (FT). If TRIP12 contains a linkage-specific UBD, it will bind to and remove the di-ubiquitin from the supernatant resulting in the loss of di-ubiquitin in the FT. In this experiment, a 6×-trypsin-resistant tandem ubiquitin-binding entity (TR-TUBE) efficiently bound and removed K11- and K48-linked di-ubiquitins as judged by the loss of specific bands in the FT by western blots, unlike TRIP12 that showed no binding to either free or linkage-specific di-ubiquitins (Supplementary Fig. 4c, d), suggesting that TRIP12 does not harbour a UBD recognising single-linkage ubiquitin chains.

To understand the mechanistic basis of TRIP12-FBW7 interaction, we generated TRIP12 deletion mutants lacking either its HECT domain or HECT/WWE domain—two well-characterised TRIP12 protein–protein interaction domains (Fig. 4h). We found that TRIP12 interaction with FBW7 requires its WWE domain (Fig. 4i). Thus, our data suggest that TRIP12 weakly interacts with FBW7 and this interaction is further enhanced by FBW7 autoubiquitylation.

**FBW7 lysines 404 and 412 are autoubiquitylated and required for TRIP12 interaction**. To better characterise how FBW7 autoubiquitylation contributes to TRIP12 interaction, we performed mass spectrometric analyses on in vitro ubiquitylated FBW7 (Fig. 5a). This analysis identified seven FBW7 lysines that were autoubiquitylated (Fig. 5a and Supplementary Fig. 5a). To assess the relevance of the autoubiquitylated lysine(s), we made multiple FBW7 deletion mutants lacking the Fbox (ΔF), the dimerisation domain (ΔD), and two WD40 propeller deletion mutants (ΔWD40-1 and ΔWD40-2) (Fig. 5b). The FBW7-ΔWD40-2 mutant that lacked all the WD40 propellers strongly stabilised FBW7 protein to an extent similar to the FBW7-ΔFbox-mutant, while the ΔWD40-1 mutant, which was only slightly larger, showed FBW7 protein stability similar to wildtype (Fig. 5c). FBW7-ΔWD40-2 was very inefficiently ubiquitylated (Supplementary Fig. 5b) despite efficient interaction with the SCF-complex protein SKP1 (Supplementary Fig. 5c). Notably, unlike the unstable mutant FBW7-ΔWD40-1, FBW7-ΔWD40-2 lacks two lysine residues (K404 and K412) identified by mass spectrometry (Fig. 5a, b and Supplementary Fig. 5a). Mutating K404 and K412 to arginines in combination, but not alone, strongly stabilised FBW7 (Fig. 5d, e). Consistent with protein stabilisation, FBW7-K404/412R showed reduced ubiquitylation (Fig. 5f) and reduced interaction with TRIP12 (Fig. 5g). The reduced TRIP12-FBW7-K404/412R interaction is most likely not due to ubiquitylation of other lysine residues on FBW7 since treatment with MLN4924, which blocks Cullin-RING ligase activity and FBW7 autoubiquitylation, also reduced FBW7 interaction with TRIP12 but did not abolish it (Fig. 4g). To further confirm this, we coimmunoprecipitated TRIP12 with wild-type FBW7, ΔFbox and ΔWD40-2, the two mutants that could not be autoubiquitylated, respectively (Supplementary Fig. 5b). Indeed, TRIP12 interacted more weakly with ΔFbox and ΔWD40-2 mutants compared to the wildtype FBW7 (Supplementary Fig. 5d). Thus, lysines K404 and K412 are required for

efficient FBW7 autoubiquitylation, facilitate TRIP12 interaction, and contribute to FBW7 proteasomal degradation.

**TRIP12-mediated K11-linked ubiquitylation is essential for FBW7 proteasomal degradation**. Next, to understand the mechanistic role of TRIP12 in FBW7 regulation, we analysed in vitro degradation of FBW7 in cell-free protein extracts (Fig. 6a). 35S-labelled FBW7 was degraded in a proteasome-dependent manner in *TRIP12*-wildtype extracts (Fig. 6b). In the presence of the proteasome inhibitor (MG132), the majority of FBW7 accumulated as high-molecular-weight FBW7$^{(UBI)n}$ (Fig. 6b). In striking contrast, in *TRIP12*-knockout extracts, the majority of FBW7 accumulated as FBW7$^{(UBI)n}$ in the absence of proteasome inhibitor treatment (Fig. 6b), suggesting that FBW7 is ubiquitylated but not degraded in the absence of *TRIP12*, and hence that TRIP12 is required for FBW7 proteasomal degradation.

To test whether TRIP12 mediates FBW7 degradation through ubiquitylation, we first identified the type of ubiquitin linkage/s involved in FBW7 proteasomal degradation. To this end, we monitored in vitro degradation of 35S-FBW7 by autoradiography in cell-free extracts in the presence of different ubiquitin mutants. As expected, in the presence of wildtype ubiquitin, FBW7 was rapidly degraded, which was partially blocked by a ubiquitin mutant that cannot be conjugated to substrates (R74; Fig. 6c). Surprisingly, in the presence of the K11R ubiquitin mutant, autoubiquitylated FBW7 accumulated as a high-molecular weight species (Fig. 6c), indicating a requirement of K11 ubiquitin linkage for efficient FBW7 protein degradation. Unexpectedly, K48 linkage was not essential for FBW7 autoubiquitylation as in presence of the K48R ubiquitin mutant FBW7 was still efficiently degraded, and the same held true for the K63R ubiquitin mutant (Fig. 6c). To better understand the ubiquitin linkage composition of autoubiquitylated FBW7, we performed ubiquitin restriction enzyme digest (Ubi-Crest) and silver staining of the resultant cleaved ubiquitin conjugates (Fig. 6d). USP2 removed the high-molecular-weight FBW7$^{(UBI)n}$ band and generated only free single ubiquitin moieties whereas the K11-specific DUB Cezanne or OTUD3 (K6- and K11-specific) had no effect (Fig. 6e, f). Strikingly, both OTUB1 (K48 specific) and AMSH (K63 specific) were able to cleave FBW7$^{(UBI)n}$, but frequently generated ubiquitin oligomer products (indicated by red stars), suggesting that FBW7 autoubiquitylation generates mixed-linkage ubiquitin chains. A Ubi-Crest experiment on polyubiquitylated FBW7 purified from *TRIP12*-knockout cells demonstrated that also in vivo autoubiquitylated FBW7 predominantly consisted of K48- and K63-linked chains (Supplementary Fig. 6a, b). This suggests that SCF$^{FBW7}$-autoubiquitylated FBW7 is comprised of a mixture of K48- and K63-linked ubiquitin conjugates.

**TRIP12 and UBE2S catalyse-branched ubiquitylation of FBW7**. A recent report suggested the role for K11-specific E2 enzyme UBE2S and a HECT-domain E3 ubiquitin ligase in building branched K11/K48 polyubiquitin chains on substrate proteins for enhanced proteasomal degradation[17]. To see whether

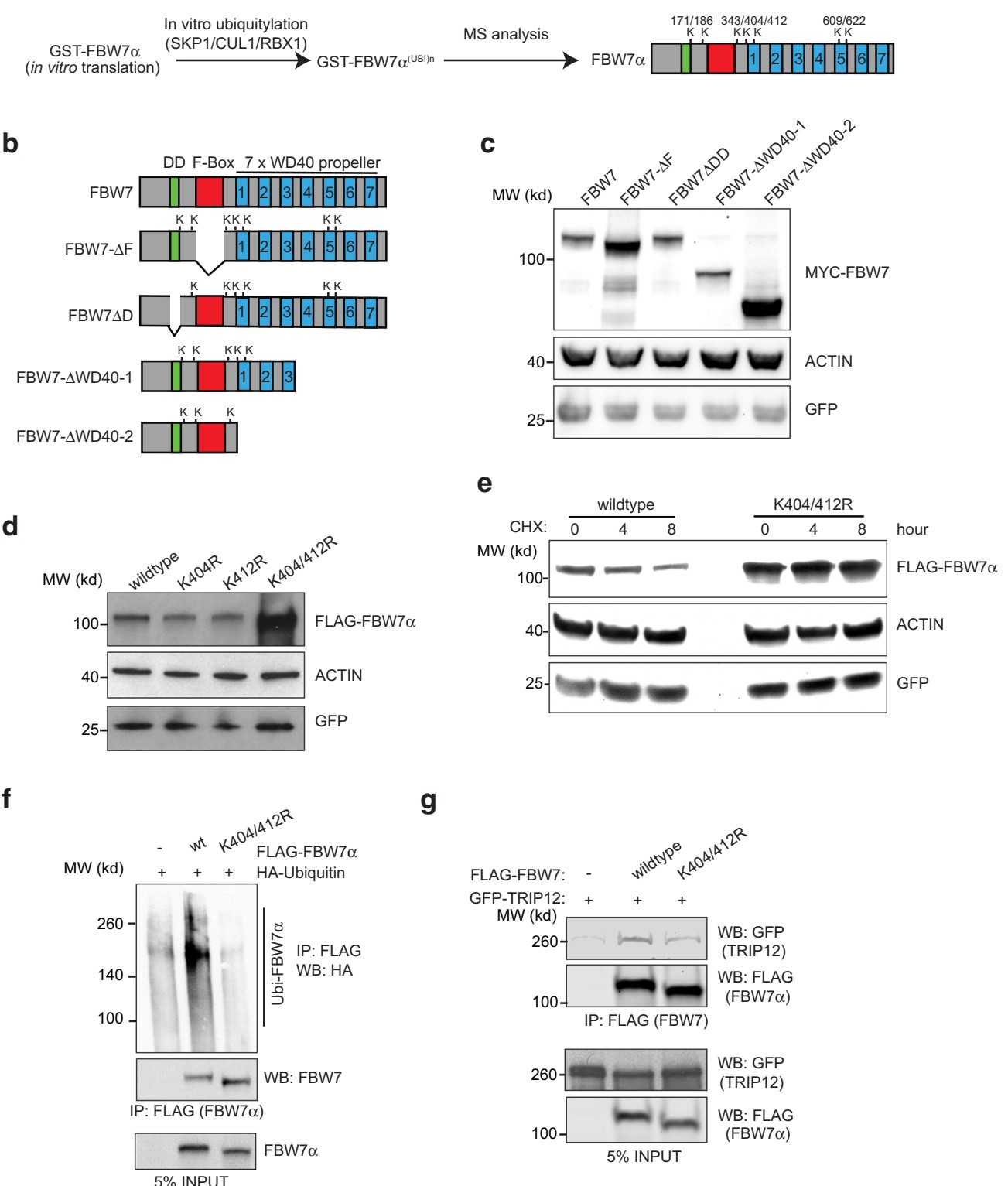

UBE2S might be the E2 required for the K11-linkage of FBW7 protein, *UBE2S* was depleted with two independent siRNAs (Supplementary Fig. 7a). *UBE2S* knockdown stabilised FBW7 protein (Fig. 7a), suggesting that UBE2S may be involved in FBW7 ubiquitylation (Fig. 7a). We next investigated whether TRIP12 could directly ubiquitylate FBW7, and whether TRIP12

was the enzyme capable of catalysing K11-linked polyubiquitin chains on FBW7 (Fig. 7b). In vitro FBW7 autoubiquitylation did not involve K11-linked polyubiquitin conjugates (Fig. 7c, lane 2), as previously suggested by the Ubi-Crest assay (Fig. 6e, f). Strikingly, addition of TRIP12 together with UBE2S catalysed K11-linked ubiquitin conjugates on autoubiquitylated FBW7 in a

**Fig. 5 FBW7 lysines 404 and 412 are autoubiquitylated and required for TRIP12 interaction. a** Schematic of in vitro GST-FBW7 ubiquitylation and mass spectrometry (MS) workflow. The cartoon shows the location of the identified ubiquitylated lysines (K) on FBW7. **b** Schematic of FBW7 deletion mutants showing ubiquitylation sites identified from experiment in **a**. **c** Western blots for MYC-FBW7 deletion mutants. GFP was used as a transfection control. **d** Western blots of the indicated FLAG-FBW7 wildtype, single, and double mutant proteins overexpressed in HEK293T cells. **e** Western blots showing the stability of the indicated FLAG-FBW7 proteins in HEK293T cells treated with 100 μg/ml cycloheximide (CHX) for the indicated time. GFP plasmid was used as transfection control and subsequently GFP and Actin blots were used as loading controls in **c–e**. **f** Western blots of ubiquitylated FLAG-FBW7 wildtype and K404/K412 mutant in HEK293T cells. **g** Western blots showing interaction of FLAG-FBW7 wildtype and K404/412 mutant with GFP-TRIP12 in HCT116 cells. All blots are representative of at least three independent experiments. See also Supplementary Fig. 5. Source data are provided as a Source Data file.

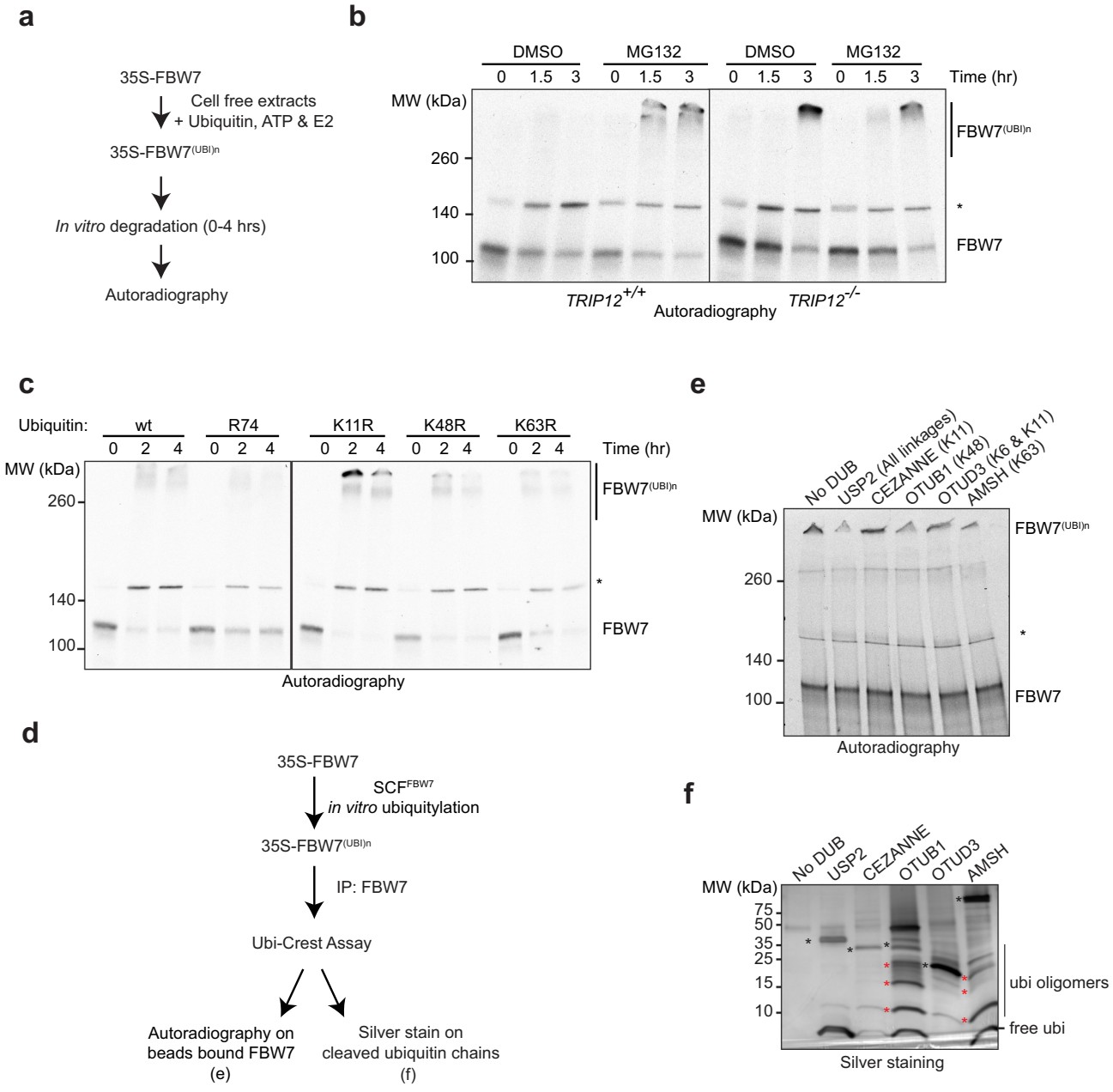

**Fig. 6 TRIP12-mediated K11-linked ubiquitylation is essential for FBW7 proteasomal degradation. a** Schematic of in vitro degradation assay. **b** In vitro 35S-FBW7α degradation assay performed in cell-free extracts from cells of the indicated genotypes and visualised by autoradiography. **c** In vitro FBW7 degradation assay performed in cell-free extracts in the presence of the indicated ubiquitin mutants and visualised by autoradiography. **d** Schematic for in vitro ubiquitylation and ubiquitin restriction enzyme digest experiment for 35S-FBW7. **e** Autoradiographs showing cleavage of beads-bound polyubiquitylated 35S-FBW7 by the indicated DUBs. Linkage specificity is given in parentheses. Black stars (*) in **b**, **c**, and **e** represent the nonspecific band of unknown identity. **f** Silver stain on cleaved ubiquitins in the supernatant from the experiment in **e**, black stars represent respective DUBs and red stars label uncleaved ubiquitin oligomers. All blots are representative of at least three independent experiments with the exception of **e**, which was done twice with similar results. Source data are provided as a Source Data file.

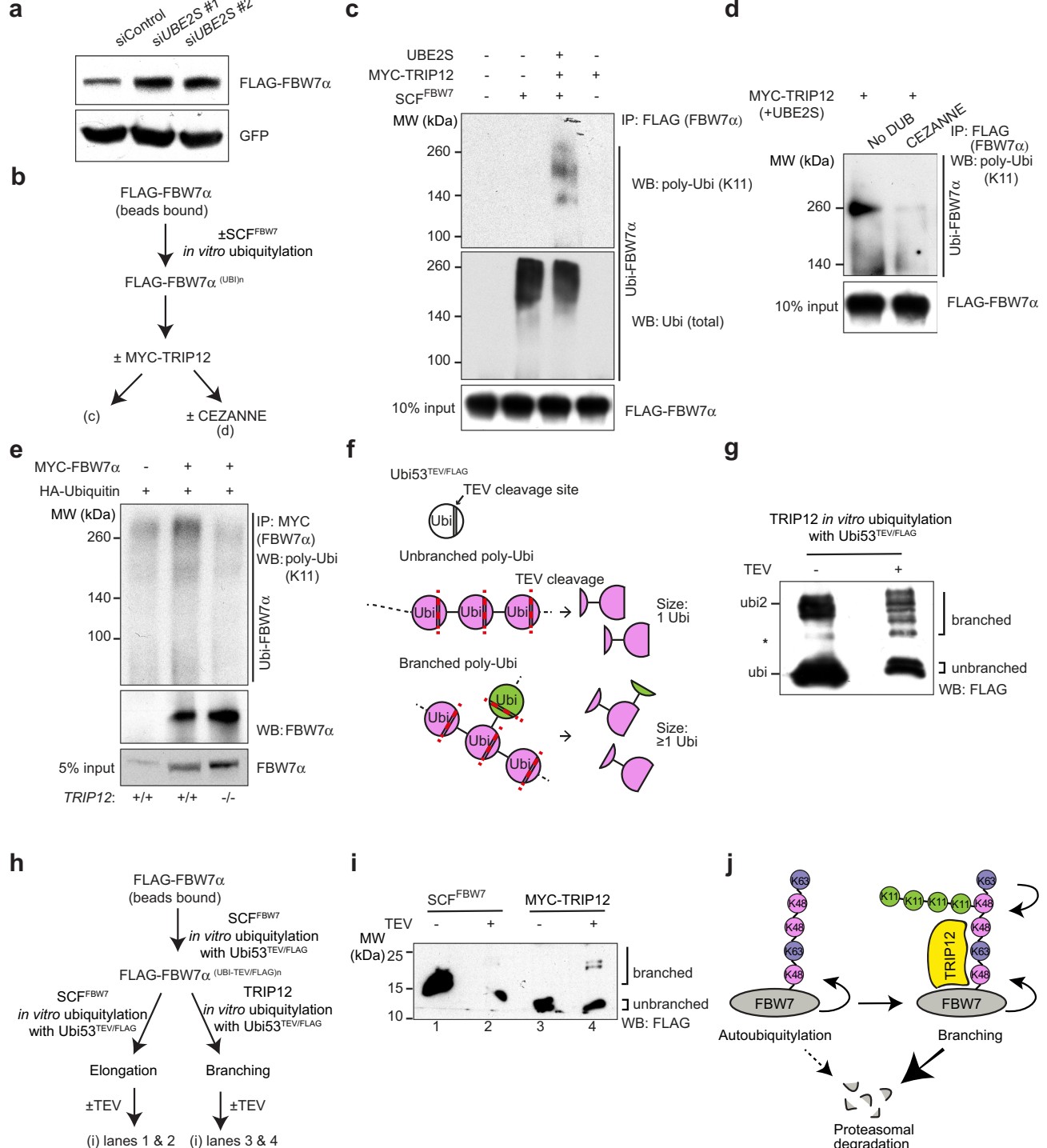

**Fig. 7 TRIP12 and UBE2S catalyse-branched ubiquitylation of FBW7. a** Western blots showing levels of FLAG-FBW7 protein in HEK293T cells incubated with a control and two independent siRNAs targeting the K11-linkage-specific E2 enzyme, *UBE2S*. **b** Schematic of sequential in vitro ubiquitylation assay followed by Cezanne-mediated K11-linkage cleavage. Ubiquitylation step 1: $SCF^{FBW7}$ + UBCH5 + wt-ubi; step 2: TRIP12 + UBE2S + K48R-ubi. **c** Western blots showing FBW7 in vitro ubiquitylation and K11-linkage addition by TRIP12. **d** Confirmation of K11-linkage on FBW7 by TRIP12 with a K11-specific deubiquitinase, Cezanne. **e** Western blots showing enrichment of K11-linked polyubiquitylated FBW7 in HEK293T cells of the indicated genotypes. **f** Outline of the method to detect branched ubiquitylation. **g** Western blots confirming branching activity of TRIP12-HECT domain in vitro, black star (*) represents a nonspecific band of unknown identity. **h** Schematic for FBW7 in vitro branched ubiquitylation assay. **i** FBW7-branched ubiquitylation by TRIP12 using the Ubi53TEV/FLAG mutant followed by TEV cleavage and blotting with FLAG-M2-HRP antibody. **j** Outline of the potential mechanism regulating FBW7 proteasomal degradation. See also Supplementary Fig. 5. All blots are representative of at least three independent experiments. Source data are provided as a Source Data file.

sequential ubiquitylation reaction, but TRIP12/UBE2S did not ubiquitylate unmodified FBW7 (Fig. 7c, lanes 3 and 4, respectively). A K11-specific DUB (Cezanne) counteracted FBW7 ubiquitylation by TRIP12 (Fig. 7b, d) and K11-linked FBW7 polyubiquitylation was reduced to background levels in *TRIP12*-knockout cells (Fig. 7e). Thus, TRIP12 catalyses the addition of K11-linked ubiquitin chains on previously autoubiquitylated FBW7.

To determine whether TRIP12 branches K11-linked conjugates on ubiquitylated FBW7 to enhance its proteasomal degradation, we used a method previously established[19], using a ubiquitin mutant with an internal tobacco etch virus (TEV) protease cleavage site along with an internal FLAG-epitope. TEV protease cleavage of unbranched UbiTEV/FLAG conjugates yields a single mono-ubiquitin band, whereas branched ubiquitylation yields higher molecular weight bands (Fig. 7f). First, we confirmed whether SCF^FBW7 can use the UbiTEV/FLAG mutant for in vitro autoubiquitylation. SCF^FBW7 efficiently autoubiquitylated FBW7 using the Ubi53TEV/FLAG mutant and as expected, TEV protease treatment led to a reduction of the high-molecular-weight polyubiquitylated species of FBW7 protein (Supplementary Fig. 7b, c). In a substrate-free reaction using a promiscuous E2 enzyme (UBCH5), the TRIP12-HECT domain robustly branched free ubiquitin (Fig. 7g). Importantly, in a sequential ubiquitylation reaction, polyubiquitin chains on autoubiquitylated FBW7 were branched by TRIP12/UBE2S (Fig. 7h, i). Thus, our data suggest that TRIP12-mediated K11-linked branched ubiquitylation of autoubiquitylated FBW7 enhances its proteasomal degradation (Fig. 7j).

## Discussion

*FBW7* is one of the most mutated ubiquitin pathway genes in human cancers with ~10–40% mutations observed in colorectal, uterine and bladder cancers[8]. Additionally, we and others have shown that FBW7 is post-translationally downregulated via different mechanisms in cancer[10,11,20]. Here, we provide the mechanistic details of FBW7 protein degradation, which is initiated by SCF^FBW7 autoubiquitylation and completed by K11-linked polyubiquitin branching by the HECT E3-ligase TRIP12.

The *yeast* homologue of TRIP12 (Ufd4) was identified as a ubiquitin fusion degradation pathway gene responsible for recognising and ubiquitylating proteins containing uncleavable N-terminal ubiquitin[12,13,21]. The role of TRIP12 in FBW7 protein regulation may have evolved from its ability to recognise and bind to ubiquitin fusion substrates. Whereas *yeast* Ufd4 extends the ubiquitin chain on a mono-ubiquitylated substrate, TRIP12 recognises polyubiquitylated FBW7, and adds a ubiquitin chain with a distinct linkage.

TRIP12 may bind to its substrates through its WWE domain as previously suggested for similar E3 ligases[22,23]. Although, we observed preferential binding of TRIP12 with polyubiquitylated FBW7 (Fig. 4a, f), TRIP12 interacted weakly with unmodified FBW7 (Fig. 5g and Supplementary Fig. 5e). Importantly, the WWE domain in TRIP12 provides substrate specificity for FBW7 since a TRIP12 mutant lacking WWE domain failed to interact with FBW7 (Fig. 4i). The interaction between TRIP12 and FBW7 is increased by autoubiquitylation. While TRIP12 does not appear to harbour a UBD that binds to single-linkage ubiquitin chains, we cannot exclude the possibility that TRIP12 contains a ubiquitin-binding domain which specifically interacts with mixed type linkages.

Current dogma holds that when FBW7 is autoubiquitylated by the SCF^FBW7 complex it is degraded by the proteasome[10]. In support of this notion, FBW7 was shown to be efficiently autoubiquitylated and a deletion mutant lacking Fbox was not degraded by the proteasome in cells[9]. However, molecular details of how SCF^FBW7 autoubiquitylation mediates FBW7 degradation were unclear.

Our analysis uncovered that SCF^FBW7 autoubiquitylation shows a marked preference for two lysines on FBW7, K404 and K412, in WD40 propeller 1 (Fig. 5d–f). Unexpectedly, we found that FBW7 autoubiquitylation by the SCF^FBW7 complex generates K48- and K63-mixed-linkage ubiquitin chains (Fig. 6e, f). K48-linked ubiquitin chains are strongly implicated in protein degradation, but in contrast, the mixed-linkage chains produced by FBW7 autoubiquitylation appear to not trigger efficient proteasomal degradation but rather serve as a priming event for subsequent TRIP12-mediated K11-linked ubiquitylation. Heterotypic K11/K48-branched ubiquitylation is known to enhance protein substrate degradation[19]. Indeed, a ubiquitin mutant that lacks K11-linkage (K11R) greatly slowed FBW7 degradation over time whereas ubiquitin mutants lacking K48 and K63 linkages (K48R and K63R) did not (Fig. 6c). Thus, FBW7 K11-linked branched ubiquitylation by TRIP12 is an important regulatory mechanism controlling FBW7 stability (Fig. 7k).

SCF^FBW7 targets several oncogenic proteins for proteasomal degradation including c-MYC, CyclinE, and MCL1. Interestingly, c-MYC protein levels were not affected in *TRIP12*-depleted cells (Fig. 3a). An important regulatory mechanism of c-MYC protein levels is nucleolar degradation by FBW7 gamma[24]. As TRIP12 only affects FBW7 alpha isoform (Supplementary Fig. 2a), we speculate that this is the reason why the effect on c-MYC protein is less pronounced. Additionally, it is also possible that once autoubiquitylated, FBW7 either dissociates from its cognate SCF-complex or preferentially interacts with some but not all substrates such that, only a subset of FBW7 substrates is downregulated upon TRIP12 depletion.

Finally, the role of FBW7 in chemotherapy resistance is well established. MCL1 is a critical FBW7 substrate that is degraded in mitosis by the SCF^FBW7 complex, and FBW7 mutations render cancer cells resistant to chemotherapeutic agents targeting the mitotic spindle such as Taxol[6,7]. Thus, there has been considerable interest in targeting MCL1 in cancers reliant on MCL1 overexpression[25–27]. Here, we show that genetic deletion of *TRIP12* led to FBW7-dependent downregulation of MCL1 protein and enhanced sensitivity to Taxol in cancer cells, phenotypes that were blocked by concomitant *FBW7* mutation or deletion (Fig. 3g, h and Supplementary Fig. 3b). Thus, cancer cells may acquire chemotherapy resistance via different mechanisms and our findings suggest TRIP12 as a potential target to tackle MCL1-driven chemotherapy resistance.

## Methods

**Plasmids**. Stephen Elledge kindly provided the GPS reporter vector[18]. FBW7α-ORF was PCR amplified with the Long template high fidelity PCR kit (#11681834001, Roche), and gateway-cloning adapters, flanking each side of the ORF, were added. A two-step gateway cloning (#11789013 & 11791100, ThermoFisher) was performed to shuttle FBW7α-ORF into the GPS vector. FBW7α deletion mutants were PCR amplified and gateway cloned into pDEST-N-MYC as explained above. Jiri Lukas (Novo Nordisk) kindly provided mammalian expression plasmids for GFP-TRIP12 and GFP-TRIP12-CA mutant. TRIP12 deletion mutants were created using primer pairs excluding the HECT or WWE domain using Gateway-cloning system (ThermoFisher). A two-step gateway cloning as above generated MYC- and FLAG- TRIP12 plasmids. Bacterial expression destination plasmids pDEST-527 and 565 were from Dominic Espinato (#11518, #11520, Addgene). CRISPR/Cas9 SAM plasmids were from Feng Zhang[28] and obtained from Addgene (#61425, #61427, and #89308). Michael Rape (University of Berkeley) kindly provided mammalian expression plasmids Ubi53TEV/FLAG and Ubi64TEV/FLAG[19]. For recombinant protein purification, UbiTEV/FLAG mutants were first PCR amplified using Ubi53TEV/FLAG and Ubi64TEV/FLAG plasmids as templates and then shuttled to pDEST-527 as described above. Validated clones were used to transform BL21(DE)-codon + *E. coli* (Agilent) for recombinant protein purification. Sequences for all the cloning primers are given in Supplementary Table 1.

**Pooled shRNA library screen.** Stable HEK293T cells were generated by transducing with the lentivirus containing the GPS-FBW7α vector. The infection was maintained at one virus/cell and the cells were selected in puromycin (1 µg/ml) for 48 h. After selection, cells were propagated, FACS sorted based on GFP expression, and a clonal population expressing uniform levels of GFP was expanded for the shRNA library screen. A lentivirus-based shRNA library targeting ~5500 human genes (DHPAC-M1-PAX, Cellecta) was packaged in to lentivirus as given in protocol. The concentrated lentivirus was taken at an MOI of 0.7, and the EGFP-FBW7α reporter cells were transduced in triplets as given in the protocol. Representation of each shRNA was maintained at least 1000 times throughout the experiment. Sorted cells ~10 million/replicate, were pelleted, DNA was harvested, and the libraries prepared for sequencing as given in the protocol. Sequencing was performed on Illumina HiSeq2500 platform.

For deconvolution of raw data, internal barcode sequences were extracted from the FastQ files by trimming each read back to the initial 18 bases using fastx_trimmer from the fastx_toolkit-0.0.14 (specifically, with parameters –f 1 -l 18 -z). These barcodes were then aligned to the Cellecta-DECIPHER-Human-Module1-HTSeq barcode library using bwa 0.6.2 with the following non-default settings: -n 1 –l 18 –k 1 –t 4. The aligned bam files were sorted and the number of reads mapping to each target were extracted using samtools sort and idxstats, respectively (samtools 0.1.16). Raw count levels were normalised by scaling to the maximum total read count across all samples (i.e. each read count was multiplied by (max_reads_aligned_all_samples/ number_reads_aligned_this_sample). These normalised read counts were log-transformed after the addition of an offset of 1 to avoid taking the log of zero. The Wilcoxon test was used to compare log-normalised read counts between samples.

**Flow cytometry.** shRNA lentivirus library transduced live DsRed-IRES-EGFP-FBW7α reporter cells were washed, trypsinized, counted, DAPI stained, re-suspended in sorting buffer (1% FBS in PBS) and filtered into single-cell suspension. DAPI positive dead cells were excluded and double-positive single scattered cells were sorted for low and high GFP/DsRed ratio (Supplementary Fig. 1c). Mock vector transduced DsRed-IRES-EGFP-FBW7α cells were used as a control for gating purpose. ~1 billion live cells/experiment were used for sorting low/high GFP-FBW7α/DsRed cells on FACS Aria II (BD Biosciences). For each population, we sorted 10 million cells/replicate. Sorted cells were pelleted and the DNA was prepared by phenol:choloroform:isoamyl alcohol method. Sorting was done on three replicates and genomic DNA was processed for sequencing library preparation and deconvolution as above.

**Validation of selected hits by FACS.** To validate library hits, two independent siRNAs (Supplementary Table 2) against each selected gene were incubated on a 12 well plate in complex with Lipofectamine RNAiMax (#13778100, ThermoFisher) with DsRed-IRES-EGFP-FBW7α reporter cell line. 48 h after the transfection, cells were washed in ice-cold PBS, trypsinized, collected in FACS buffer (0.2% FBS in PBS), filtered into single cells and DAPI stained for 20 min on ice. Stained cells were processed as above with the exception of sorting (Supplementary Fig. 1c). Analysis was done with BD Fortessa A and the data was collected by BD FACSDiva 8.0.1. Data were reanalysed in FlowJo 10 and EGFP/DsRED ratio plots were generated for protein stability of EGFP-FBW7α.

**Pathway analysis.** Pathway analysis was performed on genes that had three or more shRNAs significantly enriched in either the low or high EGFP-FBW7α populations using the Enrichr pathway analysis platform. The top five enriched pathways from KEGG 2019 are displayed with individual score and p-values.

**Site-directed mutagenesis.** FBW7α lysine to alanine mutants were made using the Quick-change lightning site-directed mutagenesis kit (Agilent). Mutated plasmid clones were validated by Sanger sequencing, and then used for subsequent overexpression in mammalian cells followed by western blot.

**Western blot and co-immunoprecipitation assays.** Immunoblots and co-immunoprecipitation (IP) were carried out as previously reported[29]. Briefly, semiconfluent cells were treated with either DMSO or a proteasome inhibitor (25 µM, MG132) for up to 4 h, washed in ice-cold PBS, trypsinized and collected in Eppendorf tubes. The cell pellets were lysed in either 1× cell lysis buffer (CLB, #9803, CST) or 9M urea buffer supplemented with 1× protease inhibitor cocktail, 1mM PMSF, 1mM NaF, and 1mM NaVO₃, and spun at 15,600 × g for 5 min. The cleared lysates were mixed with 4× laemmli SDS loading dye, boiled at 70 °C for 5 min and ran on Biorad Tris-TGX-gels (7.5 or 10%) for 1 h at 120–150 V. Proteins were then transferred to pre-cut nitrocellulose membranes (#1704271, Biorad) using the Trans-Blot Turbo Transfer System (#1704150, Biorad), blocked in 5% non-fat dry milk in 0.05% TBST and incubated with primary antibodies overnight. Membranes were washed at least 3× for 15 min each and incubated in 1:10,000 diluted anti-rabbit HRP-conjugate (#A120-101P, Abcam) for 1 h at RT, washed 3× for 15 min each and developed by chemiluminescence using ECL Prime western blotting reagent (#RPN2232, Amersham). For FBW7α and TRIP12 co-IP, MYC-FBW7 constructs were co-overexpressed with GFP- or FLAG- TRIP12 for 72 h, treated with MG132 for 4 h prior to harvesting, washed, lysed in 1X CLB, and immunoprecipitated with MYC/FLAG beads (Sigma) overnight at 4 °C. Immune

complexes were washed three times in 1× CLB, re-suspended in 4× SDS sample buffer, and run on 7.5% Tris-TGX gels (Biorad). Membranes were probed with FLAG-HRP, FBW7α and TRIP12-specific antibodies.

**Antibodies.** Antibodies used for western blotting were anti-FBW7α (#A301-720A), and anti-TRIP12, 1:500 (#301-814A) from Bethyl, anti-FLAG, 1:5000 (#M2 clone), anti-HA-HRP-conjugate, 1:5000 (#A190108P), anti-c-MYC Tag, 1:500 (#clone4A6) HRP conjugated, and anti-vinculin, 1:1000 (#V9131) from Sigma, anti-GFP, 1:1000 (#11814460001, Roche), anti-Actin HRP-conjugate, 1:10,000 (#ab-49900), anti-GAPDH, 1;1000 (#ab9485), anti-c-MYC-Y69, 1:1000 (#ab-32072), and anti-α-tubulin, 1:2000 (#ab-7291) from Abcam, anti-CUL1, 1:1000 (#71-8700, Invitrogen), anti-p19Skp1, 1:1000 (#610530, BD Biosciences), anti-CyclinE, 1:1000 (#sc-481), and anti-RNF168, 1:1000 (#sc-101125) from Santa Cruz, anti-CDKN2a, 1:1000 (#4824), phospho-p42/44 MAPK, 1:1000 (#4376), and anti-MCL1, 1:1000 (#54539) from Cell signaling, anti-conjugated ubiquitin (FK2, 1:1000) (#BML-PW8810, Enzo life sciences), and anti-Lys11 linkage-specific, 1:500 (#MABS107), was from Millipore.

**Cellular ubiquitylation assays.** 6×-His-ubiquitin and FLAG-FBW7 were co-overexpressed in stable HEK293T cell lines expressing a control or lenti-shTRIP12 (sh3′UTR). 48 h after transfection, cells were treated with MG132 for 6 h, washed, collected, and lysed in a high denaturing buffer containing 20 mM imidazole. The ubiquitylated proteins were enriched by Nickel-NTA beads for 2 h at room temperature, washed, eluted with 300 mM imidazole containing sample buffer, and resolved on 7.5% Tris-TGX gels. The blots were probed with FLAG-HRP antibody (Sigma) to visualise ubiquitylated FBW7. For ubiquitylation of MYC-FBW7α constructs, MYC-FBW7α was co-overexpressed with 6×-HA-ubiquitin in HEK293T cells for 48 h, cells were washed, collected, lysed in denaturing buffer containing 0.1% SDS, and boiled at 95 °C for 10 min. Samples were cleared by centrifugation at high speed for 10 min and diluted in 1 ml 1× CLB. Total FBW7 was immunoprecipitated with MYC beads (#E6654, Sigma) at 4 °C overnight. Immunoprecipitated proteins were blotted with FBW7α-specific antibody to visualise both unmodified and ubiquitylated FBW7, HA-HRP antibody to visualise only ubiquitylated FBW7, or with K11-linkage-specific antibody to visualise lysine 11 linkage on FBW7.

**In vitro translation.** FBW7α ORF was shuttled into pDEST-565 by a two-step gateway-cloning reaction as described above. GST-FBW7α was in vitro translated by PURExpress In vitro protein synthesis kit (#E6800, NEB). For 35S-labelled GST-FBW7α, the in vitro translation reaction mixture was coupled with 2 µl of 500 µCi EasyTag L-[35S]-Methionine (PerkinElmer).

**Recombinant protein purification.** UbiTEV/FLAG ORFs were cloned into a bacterial expression vector as described above and BL21-(DE) Codon+ E. coli were transformed. Ten colonies were picked and grown overnight at 37 °C. UbiTEV/FLAG expression was induced by 40 mM IPTG (Sigma) at room temperature. After 2–4 h of IPTG induction, a small aliquot of each colony was mixed with 4× SDS loading buffer and the extracts were boiled at 95 °C for 5 min, run on 4–20% Tris-TGX gels, and blotted with anti-FLAG-M2 HRP antibody to confirm expression of recombinant proteins. The colony giving the maximum yield of UbiTEV/FLAG was further grown into 1-l culture overnight and protein expression was induced the next morning as given above. After 4 h of IPTG induction, bacteria were pelleted at 4000×g for 30 min in 4 °C, lysed in bacterial lysis buffer (ThermoFisher), sonicated, and ultracentrifuged at 30,000×g for 2 h. The cleared lysates were either passed through a Ni-NTA column or incubated with Ni-NTA beads (Qiagen) for recombinant protein enrichment. The Ni-NTA beads were washed 6× with 10 mM imidazole containing wash buffer and eluted in 300 mM imidazole containing sample buffer. Imidazole was removed by buffer exchange column filtration at 4 °C. MYC-TRIP12 and FLAG-FBW7α were co-overexpressed in HEK293T cells and immunoprecipitated with MYC- and FLAG-M2 beads (Sigma), respectively. Beads were washed with high salt buffer (500 mM) to remove the interacting partners and then proteins were eluted by MYC and FLAG peptides in 4 °C, respectively. Protein purification was confirmed by western blotting, and 5–15 µl of purified protein was used in in vitro ubiquitylation reactions.

**Autoradiographs.** For visualising radiolabelled FBW7α, samples were boiled for 5 min in 5× SDS sample buffer at 95 °C and resolved on 7.5% Tris-TGX gels. The gels were washed in distilled water, fixed, incubated in a signal enhancer (Autofluor, National diagnostics) for 30 min, and dried on a filter paper in gel dryer for 1 h at 80 °C. The dried autoradiographs were incubated with Amersham Hyper films (GE) in gel exposure cassettes for 24–76 h before developing.

**In vitro ubiquitylation assays.** In vitro ubiquitylation was performed as previously reported[30]. Briefly, reactions were performed in the presence of 100 µM recombinant ubiquitin (U100-H), 100 nM His-UBE1 (E-304), 2.5 µM E2 conjugating enzyme (E2-616), and 10 mM ATP (B-20) all from Boston Biochem, and 1 µg recombinant SCF-complex (#23-030, Millipore) in 1× ubiquitin ligation buffer with 2–5 µl of either cold or hot GST-FBW7α as substrate. Reactions were carried out for 1 h at 37 °C, and

stopped by the addition of 5 µl of 4× SDS sample buffer and boiled at 75 °C for 5 min. Denatured samples were loaded on 7.5% Tris-TGX gels and resolved as above. For sequential ubiquitylation, in vitro translated GST-FBW7α or purified FLAG-FBW7α were first captured on anti-GST or anti-FLAG-M2 beads, washed 3× in CLB and autoubiquitylated for 1 h as described above. After 1 h, beads-bound FBW7 was washed once again in 1× ubiquitin ligation buffer and then incubated with or without purified MYC-TRIP12 (10 µl) with 5 µM UBE2S (E-690, Boston Biochem) in ubiquitin ligation mixture as given above for 3 h. Reactions were stopped by adding 4× SDS sample buffer and boiling the samples at 95 °C for 5 min. For in vitro branching of FBW7, beads-bound FLAG-FBW7α was sequentially ubiquitylated as above except that Ubi53TEV/FLAG mutant was used instead of wildtype ubiquitin. After 3 h incubation with purified TRIP12/UBE2S/UBCH5, TEV protease (Sigma) was directly added to the reaction mixture and incubated for 2 h at 34 °C. The reactions were stopped by addition of 4× SDS sample buffer and boiling the samples at 95 °C for 5 min. Samples were resolved on 4–20% Tris-Tricine gels, transferred to nitrocellulose membranes (Biorad), and blotted with FLAG-M2-HRP antibody to visualise branching. For in vitro branching by TRIP12, HECT-domain of human TRIP12 was incubated with recombinant Ubi53TEV/FLAG in ubiquitin ligation buffer as given above for 2 h at 37 °C. After 2 h, excess TEV protease was directly added to the reaction mixture for 1 h at 34 °C. Reactions were stopped by the addition of 4× SDS sample buffer and boiling at 95 °C for 5 min. The samples were resolved on 4–20% Tris-TGX gels as given above. For in vivo branching of FBW7α, MYC-FBW7α and a 1:1 mixture of Ubi53TEV/FLAG and Ubi64TEV/FLAG mutants were co-overexpressed in HEK293T cells for 48 h. After transfection, cells were treated with 25 µM MG132 for 4–6 h and processed for denaturing IP with MYC beads as given above. Beads-bound FBW7 was washed 3x in CLB and twice in TEV protease cleavage buffer. The beads-bound ubiquitylated FBW7 was then incubated with TEV protease (Sigma) for 2–4 h at 34 °C and resolved on 12% Tris-Glycine gels (Biorad) and visualised as above.

### Detection of ubiquitin-binding domains in full-length TRIP12.

FLAG-TRIP12 was overexpressed in HEK293T cells for 48 h. Cells were lysed in 1× CLB as given above and whole-cell extracts were prepared. Cleared lysates were incubated with FLAG-M2 beads (SIGMA) at 4 °C overnight. The beads-bound FLAG-TRIP12 was washed 3× in 1× ice-cold CLB supplemented with protease inhibitor cocktail (ThermoFisher). K11- and K48-linked homotypic di-ubiquitins and free mono-ubiquitin were incubated with beads-bound FLAG-TRIP12 for 2 h at 4 °C in 25 µl 25 mM Tris. In parallel, as a positive control nickel beads-bound 6x TR-TUBE (Addgene 110313) was incubated in an identical reaction exactly as above. The di-ubiquitin binding was indirectly assessed by western blotting against the remaining ubiquitins in the flow-through (FT) supernatant.

### Identification of ubiquitylated lysines on FBW7 by LC MS/MS. In vitro

translated GST-FBW7 was autoubiquitylated as above. The proteins were separated by SDS-PAGE and the high-molecular weight region was excised and sliced into 1 mm³ gel pieces. The gel pieces were de-stained with 50% acetonitrile/100 mM ammonium bicarbonate, reduced with 10 mM DTT and alkylated with 20 mM chloroacetamide. After alkylation, the proteins were digested with 300 ng trypsin overnight at 37 °C. The resulting peptides were extracted in 0.1% formic acid/2% acetonitrile and dried under vacuum centrifugation. Dried peptides were re-suspended in 100 µl of 1% TFA and cleaned using C18 Stage-tips packed in-house (one disc of C18 matrix in a 250 µl pipette tip). Peptides were washed with 1% TFA followed by elution with 80% acetonitrile/5% TFA and the eluted peptides were again dried under vacuum.

For MS analysis, peptides were re-suspended in 0.1% TFA and loaded onto 50-cm Easy Spray column (ThermoFisher). Reverse-phase chromatography was performed using the RSLC nano U3000 (ThermoFisher) with a binary buffer system at a flow rate of 250 nl/min. The nanoLC was coupled to a Q Exactive or an Orbitrap Fusion Lumos mass spectrometer using an EasySpray nano source (ThermoFisher). Both instruments were operated in data-dependent acquisition mode. The Q Exactive was acquiring HCD MS/MS scans ($R = 17,500$) after an MS1 scan ($R = 70,000$) on the 10 most abundant ions using MS1 target of $1 \times 10^6$ ions, and MS2 target of $5 \times 10^5$ ions. The maximum ion injection time utilised for MS2 scans was 120 ms, the HCD normalised collision energy was set at 28.

The Orbitrap Fusion Lumos was acquiring CID MS/MS scans after an MS1 scan ($R = 120,000$) using MS1 target of $1 \times 10^6$ ions, and MS2 target of $2 \times 10^3$ ions. The number of selected precursor ions for fragmentation was determined by the "Top Speed" acquisition algorithm and a dynamic exclusion of 60 s. The maximum ion injection time utilised for MS2 scans was 300 ms, the CID normalised collision energy was set at 35 and the ability to injections for all available parallelizable time was set to "true".

### MS data analysis. For identification of diGly containing peptides, raw data files

were analysed with MaxQuant software (version 1.6.0.13). Parent ion and tandem mass spectra were searched against Uniprot-KB Homo sapiens database. A list of 247 common laboratory contaminants provided by MaxQuant was also added to the database. For the search the enzyme specificity was set to trypsin with a maximum of three missed cleavages. The precursor mass tolerance was set to 20 ppm for the first search (used for mass re-calibration) and to 6 ppm for the main

search. Carbamidomethylation of cysteines was specified as fixed modification, oxidised methionines, N-terminal protein acetylation and di-glycine-lysine were searched as variable modifications. The datasets were filtered on posterior error probability to achieve 1% false discovery rate on protein, peptide and site level.

### In vitro deubiquitylation experiments. FLAG-FBW7 was co-overexpressed with

HA-ubiquitin for 48 h in HEK293T cells. The transfected cells were treated with MG132 as above and the ubiquitylated proteins were immunoprecipitated using HA affinity beads (A7470, Sigma). The beads-bound HA-ubiquitylated complexes were washed twice in 1× cell lysis buffer (#9803, CST) followed by 3× in PBS-Tween (0.05%) and the beads were incubated with linkage-specific deubiquitinases as given in Ubiquitin Chain Restriction (Ubi-Crest) kit (K400, Boston Biochem). The reaction was performed for 30 min at 30 °C and stopped by adding 4× sample buffer and boiling the samples at 95 °C for 5 min. The samples were resolved as above. For deubiquitylation/autoradiography, 35S-FBW7 was in vitro ubiquitylated as above, immunoprecipitated with FBW7-specific antibody overnight, and incubated with Protein Sepharose A (P3391, Sigma) for 2 h. Beads-bound FBW7 was washed as above before proceeding with Ubi-Crest experiment. Upon completion of reaction time, the supernatant from each tube with cleaved ubiquitin conjugates was transferred to a new tube, 4× sample buffer was added to all tubes, and the samples boiled at 95 °C for 5 min. The beads-bound denatured 35S-FBW7 samples were resolved on 7.5% Tris-TGX gels and processed for autoradiography as above. The cleaved ubiquitin products were run on a 4–20% Tris-TGX gel, washed, fixed, stained with silverstain as given in the kit's protocol (#LC6070, ThermoFisher), and imaged with AI600 imager (GE).

### In vitro FBW7 degradation assay. In vitro degradation assays were performed as

previously reported[31]. Briefly, 35S-labelled FBW7 was incubated with degradation cocktail in SB lysate buffer (25 mM HEPES pH = 7.5, 1.5 mM MgCl₂, 5 mM KCl, 1 mM DTT, 1× protease inhibitor cocktail (Roche), 15 mM Creatine phosphate, 2 mM ATP). Extracts were homogenised by freeze thawing and cleared by subsequent centrifugation first for 5 min at 7000×g and then at 15,600×g for 60 min. A total of 30 µl degradation cocktail was incubated with 5 µl of in vitro translated 35S-FBW7 and at indicated times, 10 µl was removed and mixed with 4 µl of 4× SDS loading buffer to stop the reaction. Upon completion of the experiment, all samples were boiled at 70–95° for 5 min, loaded on 7.5% Tris-TGX gels, and auto-radiographed as above.

### Source of cell lines. All cell lines used in this study were obtained from Cell

Services at the Francis Crick Institute, with the exception of isogeneic HCT116-*FBW7*⁺/⁺ and HCT116-*FBW7*⁻/⁻ cells which were kindly provided by Bruce Clurman (Fred Hutchinson Cancer Center, Washington). *TRIP12*-knockdown stable cell line, HEK293T cells were transduced with shRNA expressing lentivirus against a scrambled control and shRNA targeting a 3′untranslated region of the human *TRIP12* gene. Stable cells were selected by culturing in 1 µg/ml puromycin for 48 h and independent clones were picked, expanded and validated by western blotting. For generation of *TRIP12*-knockout cell lines, HEK293T and HCT116 cell lines stably expressing Cas9 protein were obtained from a high throughput screening facility of the Francis Crick Institute. The cells were plated at a density of ~40% on a 6-well plate and three independent crRNA (#CM007182-01–03, Dharmacon) targeting exon 25 of the human *TRIP12* gene were transfected in a ratio of 1:1 with tracrRNA (#U002005, Dharmacon) for 24 h. Medium was changed after 24 h, and the cells were expanded. Knockout efficiency was validated by western blot on lysates from pooled cells of each crRNA transfected cells. The line that showed maximum knockout efficiency was used for single colony isolation and propagation. For the generation of *FBW7*⁻/⁻*TRIP12*⁻/⁻ cell lines, HCT116-*FBW7*⁻/⁻ cell lines were transfected as above. The double knockout cells were colony picked, propagated, and validated as above.

**Reporting summary**. Further information on research design is available in the Nature Research Reporting Summary linked to this article.

## Data availability

All relevant data are included with the manuscript or available from the authors upon request. The raw and processed data for proteomics is uploaded on Zenodo DOI: 10.5281/zenodo.4544646 [https://zenodo.org/record/4544646#.YDleK8hKguU]. Source data are provided with this paper.

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

## Acknowledgements

We thank the Crick STPs for technical help and support. We are grateful to M. Rape, S. Elledge and J. Lukas for providing plasmids. We thank S. Urbé, M. Clague, J. Ericsson, P. Meier and K. Rittinger for the critical reading of the manuscript and C. Cremona for assistance in writing the manuscript. We thank Diego Esposito from K. Rittinger's lab at the Francis Crick Institute for the purification of recombinant Ubi53TEV/FLAG and Ubi64TEV/FLAG mutants. This work was supported by the Francis Crick Institute which receives its core funding from Cancer Research UK (FC001039), the UK Medical Research Council (FC001039), and the Wellcome Trust (FC001039). O.M.K. was funded by an EMBO long-term postdoctoral fellowship award (ALT-549-213), a Swedish International postdoctoral fellowship award (VR-537-2013-359), and by an intramural grant from Hamad Bin Khalifa University (Qatar Foundation, Qatar).

## Author contributions

Conceptualisation: O.M.K. and A.B. Formal analysis: S.H., V.E. and A.P.S. Investigation: O.M.K., J.A., J.K.N. and K.S.K. Resources: B.E.C. Writing (original draft): O.M.K. and A.B. Writing (review and editing): O.M.K., J.A., J.K.N. and A.B. Supervision: O.M.K. and A.B. Funding acquisition: A.B.

## Competing interests

The authors declare no competing interests.
