## [Peer Review File · Nature Communications]

Reviewers' Comments:

Reviewer #1:

Remarks to the Author:

This manuscript describes a novel mechanism regulating the ubiquitynation and degradation of FBW7, an F-box protein with strong implications in cancers. The authors started the study with a genetic screen for factors affecting the FBW7 protein level, and identified TRIP12, a ubiquitin ligase, as a negative regulator of FBW7. The authors further discovered that TRIP12 binds auto-ubiquitylated FBW7 and adds K-11 linked ubiquitin chains on the existing K48/K63 ubiquitin chains. This K-11 ubiquitin branch is required for the proteasome-dependent degradation of FBW7 in in vitro assays. Through regulating the degradation of FBW7, TRIP12 affects the level of FBW7 target proteins and changes the cell sensitivity to the chemotherapy Taxol. Overall, this is a nice story uncovering a novel regulatory mechanism that may play important roles in normal and diseased human cells. A variety of experiments with orthogonal approaches were performed, some of which are technically challenging, and the results are generally consistent with each other.

Besides the strength outlined above, here are some major questions:

1) The finding that FBW7 with K48/K63 linked ubiquitin chain cannot be degraded by the proteasome is novel and interesting. However, the supporting data are purely from in vitro assays. In cells with TRIP12 knockout or knockdown, whose unmodified FBW7 is increased, do they over-accumulate FBW7 with poly-ubiquitin chains (in the absence of proteasome inhibitors)? Does the over-accumulated polyubiquitylated FBW7 contain only K48/K63 ubiquitin linkages?

2) What is the substrate specificity of TRIP12? The authors mentioned, "TRIP12 preferentially interacts with ubiquitylated FBW7", but the data (Fig. 4a, 4f) suggest that only ubiquitylated FBW7 binds to TRIP12. Does TRIP12 bind any K48/K63 ubiquitin chains or does FBW7 also play a role in this interaction? This question may be answered by performing the TRIP12 co-immunoprecipitation (co-IP) experiment as shown in Fig. 4a, in the presence and absence of FBW7 knockdown, and subject the co-IP samples to Western Blot to detect poly-ubiquitin chains and chain linkage types. The co-IP samples can also be analyzed by mass spectrometry for protein IDs.

3) The interpretation of data presented in Fig. 2h is problematic. The arrow next to the Western Blot image labeled with "UM" indicates unmodified FLAG-FBW7 mutant and wild type proteins. However, this blot should detect only FLAG tagged proteins that are pulled out by Ni-NTA beads through the His6-tagged ubiquitin, so ideally, unmodified FLAG-FBW7 should not be present. It is evident that some non-specific binding occurred because FLAG signals were detected in Lane 1 and 4 where His6-ubiquitin is absent, but why did the signal increase dramatically in Lane 2 and 3 for the mutant FLAG-FBW7 that was interpreted as unmodified?

4) Based on the data presented here, FBW7 auto-ubiquitynation seems to occur in cis, because FBW7(Δ Fbox) does not seem to be ubiquitylated (Fig. 2) and the stability of FBW7(Δ D) looks similar to FBW7 (Fig. 5c). Can the authors comment on this?

Additional comments:

1) Fig. 1e: this figure will be easier to read if quantified and normalized FBW7 signals are shown. In addition, were these cells the same as illustrated in Fig. 1a? If so, was the bottom blot for RFP, not GFP?

2) Fig. 2b: typo in the x-axis and there are two "siTRIP12 #1".

3) Fig. 3c: why is a blot for GFP included?

4) Fig. 3e: it looks that these data were collected on Day 0, 5, 10, but an error bar is only visible at the last time point. How many replicates were included for each time point and what does the error bar mean?

5) Fig. 5c-e: these figures all include a blot for GFP. Is the GFP level indicating the expression activity of the tagged FBW7? If so, please state and explain.

6) Fig. 5g: FBW7(K404/412R) mutant showed reduced binding to TRIP12, consistent with the idea that TRIP12 binds ubiquitylated FBW7, and the remaining TRIP12 bound to FBW7 may be due to ubiquitylation on other Lys of FBW7. How about blocking FBW7 ubiquitylation by treating cells with the ubiquitin E1 inhibitor and then perform a similar co-IP experiment? This experiment may also tell if TRIP12 binds un-ubiquitylated FBW7.

7) Fig. 6f: what do the black and red stars mean, respectively?

8) Fig. 7c-d: on the top line, should "MYC-TRIP12" be "MYC-TRIP12+UBE2S"?

9) Methods for mass spectrometry analyses on in vitro ubiquitylated FBW7 is missing (the method for Fig. 5a and supplementary Fig. 4a).

Reviewer #2:

Remarks to the Author:

The authors seek to understand the molecular mechanism underlying how TRIP12 specifically targets Fbw7 for ubiquitination-mediated degradation in part by promoting branched K11 linked polyubiquitination. The paper is clearly written, however, the following concerns should be addressed before its publication at Nature Communications.

1. Figure 1e, it will be important to show whether depletion of TRIP12 can stabilize other Fbw7 isoform.
2. Fig. 2a, although the authors have shown that depletion of TRIP12 can prolong the half-life of ectopically expressed Fbw7, it will be nice to show endogenous Fbw7 half-life is also prolonged.
3. Fig. 3a, the authors should explain why depletion of TRIP12 can only affect the protein abundance of a subset of Fbw7 substrates, due to the difference response of Fbw7 isoform?
4. Figure 4a, it will be nice to show that delta-F Fbw7 is deficient in binding TRIP12.
5. Fig. 5g, the authors should explain or speculate why TRIP12 binds to ubiquitinated version of Fbw7. As Fbw7 undergoes mixed linkage (K48 and K63) ubiquitination, does TRIP12 contain a specific domain binding K48 linkage or K63 linkage ubiquitination chain?
6. Figure 6-7, the authors should provide bioinformatic evidence that TRIP12 is overexpressed in human cancers, thereby potentially functioning as an oncoprotein to promote the degradation of the Fbw7 tumor suppressor.

Reviewer #3:

Remarks to the Author:

The presented study by Behrens and colleagues established the E3 ubiquitin ligase TRIP12 as a novel regulator of the tumour suppressor FBW7. After an initial shRNA library screen aiming to identify regulators of FBW7 protein stability, the authors focused on TRIP12 and convincingly demonstrated that proteasomal degradation of FBW7 depends on TRIP12. Mechanistically, the authors revealed that FBW7, as part of the E3 ubiquitin ligase complex SCF, is autoubiquitinated at lysine residues K404 and K412, and that these ubiquitination events are essential for the recruitment of TRIP12, which in turn promotes K11-linked ubiquitin branching on FBW7. Furthermore, the authors showed that TRIP12

catalyzes K11-linked branched ubiquitination on FBW7 together with the E2 ubiquitin-conjugation enzyme UBE2S. Importantly, depletion of TRIP12 resulted in accumulation of FBW7 and a subsequent increase in proteasomal degradation of the SCF-FBW7 substrate MCL1, which sensitized different colorectal cancer cell lines to anti-tubulin chemotherapy.

Overall, the study describes very interesting, new and topical findings; it uncovered new mechanistic details of how the tumour suppressor FBW7 is regulated by branched ubiquitination mediated by the concerted action of two E3 ubiquitin ligases.

The quality of the work is very high, many complementary and state-of-the art approaches were used, and results are well presented in clear figures. The drawn conclusions based on the presented data are justified and do not need further validation. I can recommend publication without reservation.

Point-by-point response to reviewer comments for Khan et al. "Proteasomal degradation of the tumour suppressor FBW7 requires branched ubiquitylation by TRIP12" (NCOMMS-20-16510)

We are grateful to the editor and reviewers for assessing our manuscript and thank them for their comments. Our responses to each specific comment are in *blue italics* below.

Reviewer comments:

Reviewer #1:

1. The finding that FBW7 with K48/K63 linked ubiquitin chain cannot be degraded by the proteasome is novel and interesting. However, the supporting data are purely from in vitro assays. In cells with TRIP12 knockout or knockdown, whose unmodified FBW7 is increased, do they over-accumulate FBW7 with poly-ubiquitin chains (in the absence of proteasome inhibitors)? Does the over-accumulated polyubiquitylated FBW7 contain only K48/K63 ubiquitin linkages?

In the absence of the proteasome inhibitor, a significant proportion of the accumulated FBW7 protein in TRIP12-depleted cells is unmodified as suggested by the size of the FBW7 band migrating on the gel (Fig. 2a, 2c, and Fig S2b). We speculate that this is most likely due to counteracting deubiquitinases.

To specifically address the question raised by the reviewer, we performed His-ubiquitin FBW7 pulldown experiment in control and TRIP12-depleted cells in the absence of proteasome inhibitor MG132. Indeed, in the absence of proteasome inhibitor we find an increase in ubiquitylated FBW7 in TRIP12-depleted cells compared to wild-type controls.. This data is added in the revised manuscript (Supplementary Fig. 2c)

To check type of polyubiquitin linkages on FBW7 in TRIP12 depleted cells, we enriched polyubiquitylated FBW7 from TRIP12-knockout cells and performed Ubi-Crest experiment. In this assay, a K11-linkage specific DUB (Cezanne) was unable to cleave any poly-ubiquitin chains present on FBW7 whereas USP2, a promiscuous DUB, and the combination of OTUB1/AMSH (K48/K63 specific DUBs) – removed all ubiquitin species from FBW7 (Supplementary Figure 6a, and 6b). This experiment suggests that in the absence of TRIP12, FBW7 largely consists of K48 and K63 ubiquitin linkages and is not modified with K11-ubiquitin linkages.

2) What is the substrate specificity of TRIP12? The authors mentioned, "TRIP12 preferentially interacts with ubiquitylated FBW7", but the data (Fig. 4a, 4f) suggest that only ubiquitylated FBW7 binds to TRIP12. Does TRIP12 bind any K48/K63 ubiquitin chains or does FBW7 also play a role in this interaction? This question may be answered by performing the TRIP12 co-immunoprecipitation (co-IP) experiment as shown in Fig. 4a, in the presence and absence of FBW7 knockdown, and subject the co-IP samples to Western

Blot to detect poly-ubiquitin chains and chain linkage types. The co-IP samples can also be analyzed by mass spectrometry for protein IDs.

Here, the referee asks how TRIP12 interacts with FBW7. We have shown that the FBW7K404/412R, Δ Fbox (new data), and Δ WD40-2 (new data) mutants, which cannot be auto-ubiquitylated (Figure 5f & Supplementary Figure 5b) interact with TRIP12, although weakly (Figure 5g & Supplementary Figure 5d). Previous studies have suggested the role for WWE domain of TRIP12 for interaction with its substrates (1). Indeed, we found that deletion of the WWE domain of TRIP12 abolishes interaction with FBW7 (Figure 4h & i). This suggests that TRIP12/FBW7 interaction provides specificity, and polyubiquitin attachment strengthens this interaction, which is the typical way post-translational modifications modulate protein-protein interactions.

In addition, we have now assayed if TRIP12 can directly bind free ubiquitin chains in di-ubiquitin pulldown experiments and found that TRIP12 does not appear to bind to unanchored homotypic K48 or K11 ubiquitin linkages in isolation (Supplementary Figure 4c and 4d). This suggests that TRIP12 does not directly bind single linkage ubiquitin chains, which is consistent with requirement of direct interaction between TRIP12 and FBW7 for specificity. However, we do not rule out the possibility that TRIP12 contains a ubiquitin binding domain which specifically interacts with mixed type linkages or interacts with substrate anchored polyubiquitin chains. This is, however, near impossible to test with available reagents. In the revised manuscript we have now added a discussion of the molecular details of TRIP12/FBW7 interaction on page 15.

3) The interpretation of data presented in Fig. 2h is problematic. The arrow next to the Western Blot image labeled with "UM" indicates unmodified FLAG-FBW7 mutant and wild type proteins. However, this blot should detect only FLAG tagged proteins that are pulled out by Ni-NTA beads through the His6-tagged ubiquitin, so ideally, unmodified FLAG-FBW7 should not be present. It is evident that some non-specific binding occurred because FLAG signals were detected in Lane 1 and 4 where His6-ubiquitin is absent, but why did the signal increase dramatically in Lane 2 and 3 for the mutant FLAG-FBW7 that was interpreted as unmodified?

We agree with the reviewer that sometimes we do find nonspecific binding of unmodified (UM)-FLAG-FBW7 to Ni-NTA beads. We also agree that Fig. 2h showed unusually high unspecific binding in lane 2 & 3. We have repeated this experiment several times and one way to reduce non-specific binding of UM-FLAG-FBW7 is to increase the imidazole concentration in wash buffers. Unfortunately, this comes at a price of losing specific binding to His-ubiquitylated-FBW7. Therefore, we have now replaced the blot in question with an identical blot where we see similar levels of background between samples with and without His-Ubi. This blot is added to Fig 2h of the revised manuscript.

Minor Comments:

1) Fig. 1e: this figure will be easier to read if quantified and normalized FBW7 signals are shown. In addition, were these cells the same as illustrated in Fig. 1a? If so, was the bottom blot for RFP, not GFP?

We have now quantified the blots in Fig. 1e. The cells used in this experiment are HEK293T and a GFP expressing plasmid was used a transfection efficiency control and subsequent loading control.

2) Fig. 2b: typo in the x-axis and there are two “siTRIP12 #1”.

We thank the reviewer for spotting this mistake and have now corrected this in the updated manuscript.

3) Fig. 2c: why is a blot for GFP included?

We routinely use GFP plasmid as a transfection control in our experiments where epitope tagged FBW7 is overexpressed. The GFP blot confirms comparable transfection efficiency between samples.

4) Fig. 3e: it looks that these data were collected on Day 0, 5, 10, but an error bar is only visible at the last time point. How many replicates were included for each time point and what does the error bar mean?

The experiment in Fig. 3e was repeated 3 times and each sample was treated in duplicates. The error bars represent standard deviation. We clarify this now in the figure legend.

5) Fig. 5c-e: these figures all include a blot for GFP. Is the GFP level indicating the expression activity of the tagged FBW7? If so, please state and explain.

In Fig. 5c-e, GFP is used as a transfection control and GFP blots indicate that all samples have comparable level of transfection. We have updated the Figure legend to make this clearer.

6) Fig. 5g: FBW7(K404/412R) mutant showed reduced binding to TRIP12, consistent with the idea that TRIP12 binds ubiquitynated FBW7, and the remaining TRIP12 bound to FBW7 may be due to ubiquitynation on other Lys of FBW7. How about blocking FBW7 ubiquitynation by treating cells with the ubiquitin E1 inhibitor and then perform a similar co-IP experiment? This experiment may also tell if TRIP12 binds un-ubiquitylated FBW7.

We thank the reviewer for raising this important point. As suggested by the reviewer, we have now added new data in revised manuscript where we pretreated cells with MLN4924 (NAE1 inhibitor) to block neddylation mediated Cullin-Ring ligase activity and hence FBW7 autoubiquitylation before co-immunoprecipitation of TRIP12/FBW7. TRIP12 weakly interacted with FBW7 in the presence of MLN4924 (Figure 4g). This data is consistent with new data in Supplementary Figure 5d where TRIP12 interaction with FBW7 was reduced ~10-fold when a Δ Fbox and Δ WD40 FBW7 mutants were used.

7) Fig. 6f: what do the black and red stars mean, respectively?

We thank the reviewer for spotting the missing information in this figure. The black stars mark the size of the labelled DUBs and red stars label ubiquitin oligomers. This information is added to the figure legend of Fig. 6f.

8) Fig. 7c-d: on the top line, should “MYC-TRIP12” be “MYC-TRIP12+UBE2S”?

We thank the reviewer for spotting this mistake. We have now added the labels for UBE2S where appropriate.

9) Methods for mass spectrometry analyses on in vitro ubiquitylated FBW7 is missing (the method for Fig. 5a and supplementary Fig. 4a).

We apologise for this oversight. We have now added the missing information for Mass spectrometry experiment in 'Methods' section.

Reviewer #2:

1. Figure 1e, it will be important to show whether depletion of TRIP12 can stabilize other Fbw7 isoforms.

We have now added FBW7 isoform data in the new Supplementary Fig. 2a. We find that in TRIP12-KO cells, only FBW7 alpha is stabilised, and that we see no effect on the stability of the beta and gamma isoforms of FBW7.

2. Fig. 2a, although the authors have shown that depletion of TRIP12 can prolong the half-life of ectopically expressed Fbw7, it will be nice to show endogenous Fbw7 half-life is also prolonged.

We showed endogenous FBW7 accumulation in siTRIP12 treated HEK293 cells (Fig 2a). To address this point further, we have now added an experiment demonstrating increased stability of endogenous FBW7. IP/WB on endogenous FBW7 was done in TRIP12wt and KO cells pretreated with cycloheximide. We find a clear increase in stability of endogenous FBW7 protein in TRIP12-KO cells. The new data is shown in Supplementary Fig. 2b.

3. Fig. 3a, the authors should explain why depletion of TRIP12 can only affect the protein abundance of a subset of Fbw7 substrates, due to the difference response of Fbw7 isoform?

An important regulatory mechanism of c-Myc protein levels is nucleolar degradation by FBW7gamma (2). As TRIP12 only affects FBW7alpha, we speculate this is reason why the effect on c-Myc protein is less pronounced.

4. Figure 4a, it will be nice to show that delta-F Fbw7 is deficient in binding TRIP12.

We have now added this data and show that the $\Delta Fbox$ and $\Delta WD40$ – mutants that lack autoubiquitylation– have about 10-fold reduced binding to TRIP12 (Supplementary Figure 5d). This is consistent with the idea that autoubiquitylation enhances FBW7 binding by TRIP12.

5. Fig. 5g, the authors should explain or speculate why TRIP12 binds to ubiquitinated version of Fbw7. As Fbw7 undergoes mixed linkage (K48 and K63) ubiquitination, does TRIP12 contain a specific domain binding K48 linkage or K63 linkage ubiquitination chain?

In the revised manuscript, we show that FBW7 mutant that have greatly reduced or absent ubiquitylation, the FBW7K404/412R, $\Delta Fbox$ (new data), and $\Delta WD40-2$ (new data) mutants, interact with TRIP12, although more weakly (Figure 5g & Supplementary Fig. 5d). Previous studies have suggested the role for WWE domain of TRIP12 for interaction with its substrates (1). Indeed, we found that deletion of the WWE domain of TRIP12

abolishes interaction with FBW7 (Figure 4h & i).

In the revised manuscript, we also show that TRIP12 does not appear to bind to unanchored homotypic K48 and K11 ubiquitin linkages in isolation (Supplementary Fig. 4c & d). However, this does not rule out the possibility that TRIP12 contains a ubiquitin binding domain which specifically interacts with mixed type linkages. This is, however, near impossible to test with available reagents.

In the revised manuscript we have added a discussion of the molecular details of TRIP12/FBW7 interaction on page 15. We speculate that TRIP12/FBW7 interaction provides specificity, and polyubiquitin attachment strengthens this interaction.

Figure 6-7, the authors should provide bioinformatic evidence that TRIP12 is overexpressed in human cancers, thereby potentially functioning as an oncoprotein to promote the degradation of the Fbw7 tumor suppressor?

We thank the reviewer for raising an interesting point. We agree that if TRIP12 is required for FBW7 degradation, it may act as an oncogene by decreasing FBW7 stability. As suggested by the Reviewer we compared the

Figure 1: TRIP12 gene expression is upregulated in tumour versus adjacent normal tissues of indicated cancer types. * = $p < 0.05$

gene expression of TRIP12 in different cancers versus the corresponding normal tissues (Figure 1, <http://qepia.cancer-pku.cn/>). Indeed, several cancers (indicated by *) including Cholangiocarcinoma (CHOL), Diffuse Large B-cell Lymphoma (DLBCL), Pancreatic adenocarcinoma (PAAD), and Thymoma (THYM) showed significant upregulation of TRIP12 gene when compared to adjacent normal tissues. While this observation is interesting, it is very preliminary. Therefore, we would prefer to exclude this data from the current manuscript, but we will add this figure if the referee would find it useful.

Reviewer #3:

The presented study by Behrens and colleagues established the E3 ubiquitin ligase TRIP12 as a novel regulator of the tumour suppressor FBW7. After an initial shRNA library screen aiming to identify regulators of FBW7 protein stability, the authors focused on TRIP12 and convincingly demonstrated that

proteasomal degradation of FBW7 depends on TRIP12. Mechanistically, the authors revealed that FBW7, as part of the E3 ubiquitin ligase complex SCF, is autoubiquitinated at lysine residues K404 and K412, and that these ubiquitination events are essential for the recruitment of TRIP12, which in turn promotes K11-linked ubiquitin branching on FBW7. Furthermore, the authors showed that TRIP12 catalyzes K11-linked branched ubiquitination on FBW7 together with the E2 ubiquitin-conjugation enzyme UBE2S. Importantly, depletion of TRIP12 resulted in accumulation of FBW7 and a subsequent increase in proteasomal degradation of the SCF-FBW7 substrate MCL1, which sensitized different colorectal cancer cell lines to anti-tubulin chemotherapy. Overall, the study describes very interesting, new and topical findings; it uncovered new mechanistic details of how the tumour suppressor FBW7 is regulated by branched ubiquitination mediated by the concerted action of two E3 ubiquitin ligases. The quality of the work is very high, many complementary and state-of-the art approaches were used, and results are well presented in clear figures. The drawn conclusions based on the presented data are justified and do not need further validation. I can recommend publication without reservation.

We thank this referee for the kind comments and the support of our study.

References:

1. Gatti M, Imhof R, Huang Q, Baudis M, Altmeyer M. The Ubiquitin Ligase TRIP12 Limits PARP1 Trapping and Constrains PARP Inhibitor Efficiency. *Cell Rep.* 2020 Aug 4;32(5):107985. doi: 10.1016/j.celrep.2020.107985.
2. Markus Welcker, Amir Orian, Jonathan E Grim, Robert N Eisenman, Bruce E Clurman. A nucleolar isoform of the Fbw7 ubiquitin ligase regulates c-Myc and cell size. *Curr Biol.* 2004 Oct 26;14(20):1852-7. doi: 10.1016/j.cub.2004.09.083.

REVIEWERS' COMMENTS

Reviewer #1 (Remarks to the Author):

The authors have addressed my comments in my first review very well. Thank you and nice work! I have no further comments.

Reviewer #2 (Remarks to the Author):

The authors have addressed most of the raised concerns during this round of revision.

Reviewer #3 (Remarks to the Author):

The authors sufficiently addressed all comments and concerns raised by the reviewers, I recommend publication of the manuscript by Behrens and colleagues.